# Ollivier-Ricci Curvature for Hypergraphs: A Unified Framework

**Corinna Coupette**[1], **Sebastian Dalleiger**[1,2], **Bastian Rieck**[3,4]

[1]Max Planck Institute for Informatics
[2]CISPA Helmholtz Center for Information Security
[3]AIDOS Lab, Institute of AI for Health, Helmholtz Munich
[4]Technical University of Munich (TUM)

## Abstract

Bridging geometry and topology, curvature is a powerful and expressive invariant. While the utility of curvature has been theoretically and empirically confirmed in the context of manifolds and graphs, its generalization to the emerging domain of *hypergraphs* has remained largely unexplored. On graphs, the *Ollivier-Ricci curvature* measures differences between random walks via Wasserstein distances, thus grounding a geometric concept in ideas from probability theory and optimal transport. We develop ORCHID, a flexible framework generalizing Ollivier-Ricci curvature to hypergraphs, and prove that the resulting curvatures have favorable theoretical properties. Through extensive experiments on synthetic and real-world hypergraphs from different domains, we demonstrate that ORCHID curvatures are both scalable and useful to perform a variety of hypergraph tasks in practice.

## 1 Introduction

Hypergraphs generalize graphs by allowing any number of nodes to participate in an edge. They enable us to faithfully represent complex relations, such as co-authorship of scientific papers, multilateral interactions between chemicals, or group conversations, which cannot be adequately captured by graphs. While hypergraphs are more expressive than graphs and other relational objects like simplicial complexes, they are harder to analyze both theoretically and empirically, and many concepts that have proven useful for understanding graphs have yet to be transferred to the hypergraph setting.

*Curvature* has established itself as a powerful characteristic of Riemannian manifolds, as it permits the description of *global properties* through *local measurements* by harmonizing ideas from geometry and topology. For graphs, *graph curvature* measures to what extent the neighborhood of an edge deviates from certain idealized model spaces, such as cliques, grids, or trees. It has proven helpful, for example, in assessing differences between real-world networks (Samal et al., 2018), identifying bottlenecks in real-world networks (Gosztolai & Arnaudon, 2021), and alleviating oversquashing in graph neural networks (Topping et al., 2022). One prominent notion of graph curvature is *Ollivier-Ricci curvature* (ORC). ORC compares random walks based at specific nodes, revealing differences in the information diffusion behavior in the graph. As the sizes of edges and edge intersections can vary in hypergraphs, there are many ways to generalize ORC to hypergraphs. While some notions of hypergraph ORC have been previously studied in isolation (e.g., Asoodeh et al., 2018; Eidi & Jost, 2020; Leal et al., 2020), a unified framework for their definition and computation is still lacking.

**Contributions.** We introduce ORCHID, a unified framework for Ollivier-Ricci curvature on hypergraphs. ORCHID integrates and generalizes existing approaches to hypergraph ORC. Our work is the first to identify the individual building blocks shared by all notions of hypergraph ORC, and to perform a rigorous theoretical and empirical analysis of the resulting curvature formulations. We develop hypergraph ORC notions that are aligned with our geometric intuition while still efficient to compute, and we demonstrate the utility of these notions in practice through extensive experiments.

**Structure.** After providing the necessary background on graphs and hypergraphs and recalling the definition of Ollivier-Ricci curvature for graphs in Section 2, we introduce ORCHID, our framework for hypergraph ORC, and analyze the theoretical properties of ORCHID curvatures in Section 3. We assess the empirical properties and practical utility of ORCHID curvatures through extensive experiments in Section 4, and discuss limitations and potential extensions of ORCHID as well as directions for future work in Section 5. Further materials are provided in Appendices A.1 to A.5.

## 2 PRELIMINARIES

**Graphs and Hypergraphs** A *simple graph* $G = (V, E)$ is a tuple containing $n$ nodes (vertices) $V = \{v_1, \ldots, v_n\}$ and $m$ edges $E = \{e_1, \ldots, e_m\}$, with $e_i \in \binom{V}{2}$ for all $i \in [m]$. Here, for a set $S$ and a positive integer $k \leq |S|$, $\binom{S}{k}$ denotes the set of all $k$-element subsets of $S$, and for $x \in \mathbb{N}$ with $0 \notin \mathbb{N}$, $[x] = \{i \in \mathbb{N} \mid i \leq x\}$. In *multi-graphs*, edges can occur multiple times, and hence, $E = (e_1, \ldots, e_m)$ is an indexed family of sets, with $e_i \in \binom{V}{2}$ for all $i \in [m]$. Generalizing simple graphs, a *simple hypergraph* $H = (V, E)$ is a tuple containing $n$ nodes $V$ and $m$ hyperedges $E \subseteq \mathcal{P}(V) \setminus \emptyset$, i.e., in contrast to edges, hyperedges can have any cardinality $r \in [n]$. In a *multi-hypergraph*, $E = (e_1, \ldots, e_m)$ is an indexed family of sets, with $e_i \subseteq V$ for all $i \in [m]$. We assume that all our hypergraphs are multi-hypergraphs, and we drop the prefix *hyper* from *hypergraph* and *hyperedge* where it is clear from context.

We denote the degree of node $i$, i.e., the number of edges containing $i$, by $\deg(i) = |\{e \in E \mid i \in e\}|$, write $i \sim j$ if $i$ is adjacent to $j$ (i.e., there exists $e \in E$ such that $\{i, j\} \subseteq e$), and use $\mathcal{N}(i)$ ($\mathcal{N}(e)$) for the neighborhood of $i$ ($e$), i.e., the set of nodes adjacent to $i$ (edges intersecting edge $e$). While $\deg(i) = |\mathcal{N}(i)|$ in simple graphs and $\deg(i) \geq |\mathcal{N}(i)|$ in multigraphs, these relations do not generally hold for hypergraphs. Two nodes $i \neq j$ are *connected* in $H$ if there is a sequence of nodes $i = v_1, v_2, \ldots, v_{k-1}, v_k = j$ such that $v_l \sim v_{l+1}$ for all $l \in [k]$. Every such sequence is a *path* in $H$, whose *length* is the cardinality of the set of edges used in the adjacency relation. We refer to the length of a shortest path connecting nodes $i, j$ as the *distance* between them, denoted as $\mathrm{d}(i, j)$. We assume that all (hyper)graphs are *connected*, i.e., there exists a path between all pairs of nodes. This turns $H$ into a metric space $(H, \mathrm{d})$ with *diameter* $\mathrm{diam}(H) := \max\{\mathrm{d}(i, j) \mid i, j \in V\}$.

(Hyper)graphs in which all nodes have the same degree $k$ ($\deg(i) = k$ for all $i \in V$) are called $k$-*regular*. Three properties of hypergraphs that distinguish them from graphs give rise to additional (ir)regularities. First, *hyperedges* can vary in cardinality, and a hypergraph in which all hyperedges have the same cardinality $r$ ($|e| = r$ for all $e \in E$) is called $r$-*uniform*. Second, *hyperedge intersections* can have cardinality greater than $1$, and we call a hypergraph $s$-*intersecting* if all nonempty edge intersections have the same cardinality $s$ ($e \cap f \neq \emptyset \Leftrightarrow |e \cap f| = s$ for all $e, f \in E$). Third, nodes can *cooccur in any number of hyperedges*; we call a hypergraph $c$-*cooccurrent* if each node cooccurs $c$ times with any of its neighbors ($i \sim j \Leftrightarrow |\{e \in E \mid \{i, j\} \subseteq e\}| = c$ for all $i, j \in V$). Using this terminology, simple graphs are $2$-uniform, $1$-intersecting, $1$-cooccurrent hypergraphs.

Given a hypergraph $H = (V, E)$, the *unweighted clique expansion* of $H$ is $G^\circ = (V, E^\circ)$ with $E^\circ = \{\{i, j\} \mid \{i, j\} \subseteq e \text{ for some } e \in E\}$, where two nodes are adjacent in $G^\circ$ if and only if they are adjacent in $H$. The *weighted clique expansion* of $H$ is $G^\circ$ endowed with a weighting function $w \colon E^\circ \to \mathbb{N}$, where $w(e) = |\{e \in E \mid \{i, j\} \subseteq e\}|$ for each $e \in E^\circ$, i.e., an edge $\{i, j\}$ is weighted by how often $i$ and $j$ cooccur in edges from $H$. Both of these transformations are lossy, i.e., we cannot uniquely reconstruct $H$ from $G^\circ$. The *unweighted star expansion* of $H$ is the bipartite graph $G' = (V', E')$ with $V' = V \dot\cup E$ and $E' = \{\{i, e\} \mid i \in V, e \in E, i \in e\}$, and we can uniquely reconstruct $H$ from $G'$ if we know which of its parts corresponds to the original node set of $H$.

**Ollivier-Ricci Curvature for Graphs** Ollivier-Ricci curvature (ORC) extends the notion of Ricci curvature, defined for Riemannian manifolds, to metric spaces equipped with a probability measure or, equivalently, a random walk (Ollivier, 2007; 2009). On graphs, which are metric spaces with the shortest-path distance $\mathrm{d}(\cdot, \cdot)$, the ORC $\kappa$ of a pair of nodes $\{i, j\}$ is defined as

$$\kappa(i, j) := 1 - \frac{1}{\mathrm{d}(i, j)} \, \mathrm{W}_1(\mu_i, \mu_j) \,, \text{ and hence, } \kappa(i, j) = 1 - \mathrm{W}_1(\mu_i, \mu_j) \text{ if } i \sim j \,, \quad (1)$$

where $\mu_i$ is a probability measure associated with node $i$ that depends measurably on $i$ and has finite first moment, and $\mathrm{W}_1$ is the *Wasserstein distance* of order 1, which captures the amount of

work needed to transport the probability mass from $\mu_i$ to $\mu_j$ in an optimal coupling. The use of the shortest-path distance is necessary to ensure that ORC is also well-defined for pairs of non-adjacent nodes. This definition on edges or pairs of nodes alludes to the fact that Ricci curvature is associated to tangent vectors of a manifold. A common strategy to measure curvature at a node $i$ is to average over the curvatures of all edges incident with $i$ (Banerjee, 2021; Jost & Liu, 2014), i.e.,

$$\kappa(i) = \frac{1}{\deg(i)} \sum_{\{i,j\} \in E} \kappa(i,j) \,. \tag{2}$$

A popular probability measure that easily generalizes to weighted graphs and multigraphs is

$$\mu_i^\alpha(j) := \begin{cases} \alpha & j = i \\ (1-\alpha)\frac{1}{\deg(i)} & i \sim j \\ 0 & \text{otherwise} \,, \end{cases} \tag{3}$$

where $\alpha$ serves as a smoothing parameter (Lin et al., 2011). With this definition, stacking the probability measures yields the transition matrix of an $\alpha$-lazy random walk.

## 3 THEORY

Having introduced the concept of hypergraphs and the definition of Ollivier-Ricci curvature (ORC) for graphs, we now develop our framework for ORC on hypergraphs, called ORCHID (Ollivier-Ricci Curvature for Hypergraphs In Data). We focus our exposition on undirected, unweighted multi-hypergraphs, but ORCHID straightforwardly generalizes to other hypergraph variants.

### 3.1 OLLIVIER-RICCI CURVATURES FOR HYPERGRAPHS (ORCHID CURVATURES)

As mentioned in Section 2, hypergraphs differ from graphs in that edges can have any cardinality, and consequently, edges can intersect in more than one node, and nodes can co-occur in more than one edge. When generalizing ORC as defined in Section 2 to hypergraphs, these peculiarities become relevant in two places: (1) in the generalization of the measure $\mu$ for nodes, and (2) in the generalization of the distance metric $\mathrm{W}_1$. Construing the distance metric as a function *aggregating* measures (AGG), with AGG: $V^+ \to \mathbb{R}$, we can rewrite Eq. (1) for pairs of nodes $\{i,j\}$ as

$$\kappa(i,j) := 1 - \frac{\mathrm{AGG}(\mu_i, \mu_j)}{\mathrm{d}(i,j)} \,, \tag{4}$$

which facilitates its generalization; we will also use $\kappa(e)$ for (hyper)edges as a shorthand notation for Eq. (4). When defining probability measures and AGG functions on hypergraphs, we would like to retain as much flexibility as possible while also ensuring the following conditions:

 I. *Mathematical generalization.* For graphs, AGG simplifies to the original ORC on graphs.

 II. *Permutation invariance.* AGG$(e)$ = AGG$(\sigma(e))$ for edges $e$ and all node index permutations $\sigma$.

 III. *Scalability.* The probability measures and AGG functions should be efficiently computable.

Beyond these properties, we would also like to have the following *interpretability* features to ascertain that a hypergraph curvature measure is a *conceptual generalization* of ORC:

 A. *Probabilistic intuition.* The probability measures assigned to nodes should correspond to a semantically sensible random walk on the hypergraph.

 B. *Optimal transport intuition.* The generalization of the distance metric (AGG) should have a semantically sensible interpretation in terms of optimal transport.

 C. *Geometric intuition.* Edges in hypercliques should have positive curvature, edges in hypergrids should have curvature zero, and edges in hypertrees should have negative curvature.

We now specify probability measures and AGG functions for which the conditions above hold.

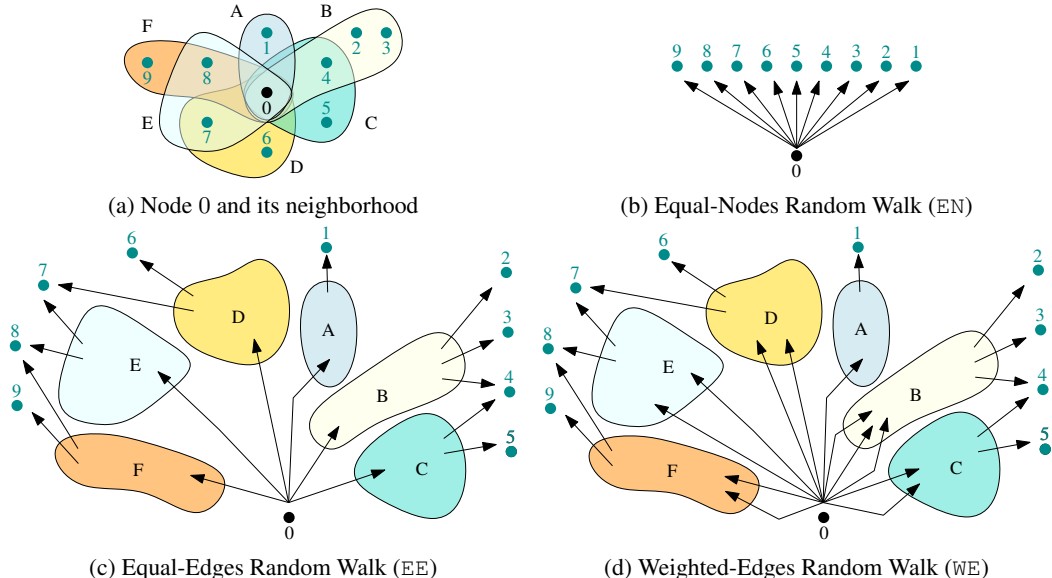

Figure 1: ORCHID's probability measures are based on random walks, depicted for the neighborhood of a node 0. Arrows outgoing from the same node or edge are traversed with uniform probability.

**Probability Measures ($\mu$).** In graphs, the most natural probability measures are induced by the $\alpha$-lazy random walk given in Eq. (3): With probability $\alpha$, we stay at the current node $i$, and with probability $(1-\alpha)/\deg(i)$, we move to one of its neighbors. There are at least three direct extensions of this formulation to hypergraphs that all retain this probabilistic intuition, thus fulfilling the requirement of Feature A. These extensions, illustrated in Fig. 1, differ only in how they distribute the $(1-\alpha)$ probability mass in Eq. (3) from node $i$ to the nodes in $i$'s neighborhood. Given a hypergraph $H$, for $i$ and $j$ with $i \sim j$, first, we could define

$$\mu_i^{\text{EN}}(j) := (1 - \alpha) \frac{1}{|\mathcal{N}(i)|} \, , \tag{5}$$

by which we pick a neighbor $j$ of node $i$ uniformly at random. We call this the *equal-nodes random walk* (EN), which is a random walk on the *unweighted clique expansion* of $H$. Second, we could set

$$\mu_i^{\text{EE}}(j) := (1 - \alpha) \frac{1}{\deg(i) - |\{e \ni i \mid |e| = 1\}|} \sum_{e \supseteq \{i,j\}} \frac{1}{|e| - 1} \, , \tag{6}$$

which first picks an edge $e \ni i$ with $|e| \geq 2$, then picks a node $j \in e \setminus \{i\}$, both uniformly at random. We call this the *equal-edges random walk* (EE), which is a two-step random walk on the *unweighted star expansion* of $H$, starting at a node $i \in V$, and non-backtracking in the second step. It underlies the curvatures studied by Asoodeh et al. (2018) and Banerjee (2021). Third, we could define

$$\mu_i^{\text{WE}}(j) := (1 - \alpha) \sum_{e \supseteq \{i,j\}} \frac{|e| - 1}{\sum_{f \ni i} (|f| - 1)} \frac{1}{|e| - 1} = (1 - \alpha) \frac{|\{e \in E \mid \{i,j\} \subseteq e\}|}{\sum_{f \ni i} (|f| - 1)} \, , \tag{7}$$

first picking an edge $e$ incident with $i$ with probability proportional to its cardinality, then picking a node $j \in e \setminus \{i\}$ uniformly at random. We call this the *weighted-edges random walk* (WE): a two-step random walk from a node $i \in V$ on a specific *directed weighted star expansion* of $H$ whose second step is non-backtracking—or equivalently, a random walk on a *weighted clique expansion* of $H$.

**Similarity Measures (AGG).** In the original formulation of ORC, i.e., Eq. (1), when determining the curvature of an edge $\{i, j\}$, the Wasserstein distance $W_1$ is used to aggregate the probability measures of $i$ and $j$. There are at least three different extensions of this aggregation scheme to hypergraphs that retain an optimal transport intuition, as required by Feature B. Leveraging that an edge $e \subseteq V$ is simply a set of nodes, the easiest extension is to leave the aggregation function

unchanged. We continue determining the curvature for pairs of nodes, and account for the edges in $H$ only in the definition of our probability measure. In this case, we could derive a curvature for an edge $e$ as the average over all curvatures of node pairs contained in $e$, i.e., we could define $\text{AGG}$ as

$$\text{AGG}_\text{A}(e) := \frac{2}{|e|(|e|-1)} \sum_{\{i,j\} \subseteq e} \text{W}_1(\mu_i, \mu_j) \,. \tag{8}$$

This is equivalent to computing the curvature of $e$ based on the average over all $\text{W}_1$ distances of probability measures associated with nodes contained in $e$:

$$\kappa_\text{A}(e) := 1 - \text{AGG}_\text{A}(e) = 1 - \frac{2}{|e|(|e|-1)} \sum_{\{i,j\} \subseteq e} \text{W}_1(\mu_i, \mu_j) = \frac{2}{|e|(|e|-1)} \sum_{\{i,j\} \subseteq e} \kappa(i,j) \,. \tag{9}$$

Intuitively, this definition assesses the mean amount of work needed to transport the probability mass from one node in $e$ to another node in $e$. Alternatively, and still keeping with the intuition from optimal transport, we can define $\text{AGG}$ as

$$\text{AGG}_\text{B}(e) := \frac{1}{|e|-1} \sum_{i \in e} \text{W}_1(\mu_i, \bar{\mu}) \,, \quad \text{and consequently,} \quad \kappa_\text{B}(e) := 1 - \text{AGG}_\text{B}(e) \,, \tag{10}$$

where $\bar{\mu}$ denotes the Wasserstein barycenter of the probability measures of nodes contained in $e$, and the denominator generalizes the original $\text{d}(i,j)$. Asoodeh et al. (2018) use this aggregation function. Intuitively, $\text{AGG}_\text{B}$ is proportional to the minimum amount of work needed to transport all probability mass from the probability measures of the nodes to one place, with the caveat that this place need not correspond to a node in the underlying hypergraph. Finally, we can capture the maximum amount of work needed to transport all probability mass from one node in $e$ to another node in $e$ as

$$\text{AGG}_\text{M}(e) := \max\{\text{W}_1(\mu_i, \mu_j) \mid \{i,j\} \subseteq e\} \,, \quad \text{and consequently,} \quad \kappa_\text{M}(e) := 1 - \text{AGG}_\text{M}(e) \,. \tag{11}$$

Independent of the choice of $\text{AGG}$, the curvature at a node $i$ can be defined as the mean of all curvatures of meaningful directions containing $i$, i.e.,

$$\kappa^\mathcal{N}(i) := \frac{1}{|\mathcal{N}(i)|} \sum_{j \in \mathcal{N}(i)} \kappa(i,j) \,, \tag{12}$$

or it can be derived as the mean of all curvatures of edges containing $i$, i.e.,

$$\kappa^E(i) := \frac{1}{\deg(i)} \sum_{e \ni i} \kappa(e) \,. \tag{13}$$

Finally, since $H$ is connected, we can define the curvature of an arbitrary subset of nodes $s \subseteq V$ as

$$\kappa(s) := 1 - \frac{\text{AGG}(s)}{\text{d}(s)} \,, \tag{14}$$

where $\text{AGG}$ can be any of our aggregation functions, and $\text{d}(s) := \max\{\text{d}(i,j) \mid \{i,j\} \subseteq s\}$ refers to the *extent* of the subset $s$. Note that for $s \in E$, $\text{d}(s) = 1$, and thus, Eq. (14) is consistent with our previous definitions of hyperedge curvatures.

## 3.2 Properties of Orchid Curvatures

Having introduced our probability measures ($\mu$) and aggregation functions ($\text{AGG}$), we now analyze their properties and the properties of the resulting curvatures. All proofs are deferred to Appendix A.1. First, we note that $\mu^\text{EN}$, $\mu^\text{EE}$, and $\mu^\text{WE}$ are equivalent for certain hypergraph classes, and all aggregation functions coincide for graphs.

**Lemma 1.** *For graphs and $r$-uniform, $k$-regular, $c$-cooccurrent hypergraphs, $\mu^\text{EN} = \mu^\text{EE} = \mu^\text{WE}$.*

**Lemma 2.** *For graphs, i.e., 2-uniform hypergraphs, we have $\text{AGG}_\text{A}(e) = \text{AGG}_\text{B}(e) = \text{AGG}_\text{M}(e)$ for all edges $e \in E$.*

Taken together, Lemma 1 and Lemma 2 imply that for graphs, ORCHID simplifies to ORC, regardless of the choice of probability measure and aggregation function. This fulfills Condition I. Moreover, *all* our aggregation functions are permutation-invariant by construction, thus satisfying Condition II. Concerning Condition III, $\kappa_{\text{A}}$ and $\kappa_{\text{M}}$ exhibit better scalability than $\kappa_{\text{B}}$, as Wasserstein barycenters are harder to compute than individual distances (Cuturi & Doucet, 2014). Another reason to prefer $\kappa_{\text{A}}$ and $\kappa_{\text{M}}$ over $\kappa_{\text{B}}$ is the existence of upper and lower bounds that are easy to calculate. To this end, let $\mathrm{d}_{\min}(H) \coloneqq \min\{\mathrm{d}(u,v) \mid u \neq v \in V\}$ be the smallest nonzero distance in $H$, and let $\|\cdot\|_1$ refer to the $L_1$ norm of a vector. We then obtain the following bounds for $\kappa_{\text{A}}$ and $\kappa_{\text{M}}$.

**Theorem 3.** *For any probability measure $\mu$ and $C(e) \coloneqq {}^{2}/_{|e|(|e|-1)}$, the curvature $\kappa_{\text{A}}(e)$ of an edge $e \in E$ is bounded by*

$$1 - \mathrm{diam}(H)C(e) \sum_{\{i,j\} \subseteq e} \|\mu_i - \mu_j\|_1 \leq \kappa_{\text{A}}(e) \leq 1 - \mathrm{d}_{\min}(H)C(e) \sum_{\{i,j\} \subseteq e} \|\mu_i - \mu_j\|_1 . \quad (15)$$

**Theorem 4.** *For any probability measure $\mu$, the curvature $\kappa_{\text{M}}(e)$ of an edge $e \in E$ is bounded by*

$$1 - \mathrm{diam}(H) \max_{\{i,j\} \subseteq e} \|\mu_i - \mu_j\|_1 \leq \kappa_{\text{M}}(e) \leq 1 - \mathrm{d}_{\min}(H) \max_{\{i,j\} \subseteq e} \|\mu_i - \mu_j\|_1 . \quad (16)$$

Directly from our definitions, we further obtain the following relationships between $\kappa_{\text{A}}$, $\kappa_{\text{B}}$, and $\kappa_{\text{M}}$, and between ORCHID curvatures on hypergraphs and ORC on their unweighted clique expansions.

**Corollary 5.** *Given a hypergraph $H = (V, E)$, $\kappa_{\text{M}}(e) \leq \kappa_{\text{A}}(e)$ and $\kappa_{\text{M}}(e) \leq \kappa_{\text{B}}(e)$ for all $e \in E$.*

**Corollary 6.** *Given a hypergraph $H = (V, E)$ and its unweighted clique expansion $G^{\circ} = (V, E^{\circ})$, for $\{i,j\} \in E^{\circ}$, the ORC $\kappa(i,j)$ in $G^{\circ}$ equals its ORCHID curvature $\kappa(i,j)$ of direction $\{i,j\} \subseteq V$ in $H$ with $\mu^{\text{EN}}$, and the ORC $\kappa(i)$ of $i \in V$ in $G^{\circ}$ equals its ORCHID curvature $\kappa^{\mathcal{N}}(i)$ in $H$ with $\mu^{\text{EN}}$.*

Corollary 6 clarifies that the equal-nodes random walk establishes the connection between ORCHID and ORC on graphs. Moreover, ORCHID curvatures capture relations between *global* properties and *local* measurements, similar to the Bonnet–Myers theorem in Riemannian geometry (Myers, 1941).

**Theorem 7.** *Given a subset of nodes $s \subseteq V$ and an arbitrary probability measure $\mu$, let $\delta_i$ denote a Dirac measure at node $i$, and let $\mathrm{J}(\mu_i) \coloneqq \mathrm{W}_1(\delta_i, \mu_i)$ denote the* jump probability *of $\mu_i$. If (i) all curvatures based on $\mu$ are strictly positive, i.e., $\kappa(s) \geq \kappa > 0$ for all $s \subseteq V$, and (ii) $\mathrm{W}_1(\mu_i, \mu_j) \leq \text{AGG}(s)$ for $\{i,j\} = \mathrm{argmax}(\mathrm{d}(s))$, then*

$$\mathrm{d}(s) \leq \frac{\mathrm{J}(i) + \mathrm{J}(j)}{\kappa(s)} . \quad (17)$$

Note that condition (ii) of Theorem 7 is always satisfied by $\text{AGG}_{\text{M}}$. Finally, in Appendix A.1, we generalize the concepts of cliques, grids, and trees (prototypical positively curved, flat, and negatively curved graphs) to hypergraphs, and we prove the following lemmas to ensure that ORCHID curvatures respect our geometric intuition, as required by Feature C.

**Theorem 8** (Hyperclique curvature). *For an edge $e$ in a hyperclique $H = (V, E)$ on $n$ nodes with edges $E = \binom{V}{r}$ for some $r \leq n$, with $\alpha = 0$,*

$$\kappa(e) = 1 - \frac{1}{n-1}, \ \textit{i.e.,} \ \lim_{n \to \infty} \kappa(e) = 1, \ \textit{independent of } r.$$

**Theorem 9** (Hypergrid curvature). *For an edge $e$ in a $r$-uniform, $k$-regular hypergrid, with $\alpha = 0$, $\kappa(e) = 0$, independent of $r$ and $k$.*

**Theorem 10** (Hypertree curvature). *For an edge $e$ in a $r$-uniform, $k$-regular, $1$-intersecting hypertree,*

$$\textit{with } \alpha = 0, \ \kappa(e) = 1 - \left( \frac{3(k-1)}{k} + \frac{1}{(r-1)k} \right), \ \textit{i.e.,} \ \lim_{k \to \infty} \kappa(e) = -2, \ \textit{independent of } r.$$

Table 1: Hypergraphs used in ORCHID experiments cover several domains and orders of magnitude. $n$ and $m$ are node and edge counts, $n/m$ is the aspect ratio, $c$ is the number of filled cells in the node-to-edge incidence matrix, $c/nm$ is the density, and $N$ is the number of hypergraphs in a collection.

(a) Individual Hypergraphs

|  | Nodes | Edges | $n$ | $m$ | $n/m$ | $c$ | $c/nm$ |
|---|---|---|---|---|---|---|---|
| aps-a | Authors | APS Papers | 505 827 | 688 707 | 0.7345 | 2 480 373 | 0.000007 |
| dblp | Authors | DBLP Papers | 3 108 658 | 6 011 388 | 0.5171 | 19 411 479 | 0.000001 |
| ndc-ai | Active Ingr. | NDC Drugs | 7 090 | 131 450 | 0.0539 | 224 084 | 0.000240 |
| ndc-pc | Pharm. Classes | NDC Drugs | 1 263 | 70 101 | 0.0180 | 273 088 | 0.003084 |

(b) Hypergraph Collections

|  | Nodes | Edges | Graphs | $N$ | $(n/m)_{\max}$ | $(c/nm)_{\max}$ |
|---|---|---|---|---|---|---|
| aps-av | Authors | APS Papers | Journals | 19 | 4.698182 | 0.005216 |
| aps-cv | APS Cited P. | APS Citing P. | Journals | 19 | 1.396552 | 0.028430 |
| dblp-v | Authors | DBLP Papers | (Groups of) Venues | 1 193 | 5.599424 | 0.002443 |
| mus | Frequencies | Chords | Music Pieces | 1 944 | 1.454545 | 0.375000 |
| stex | Tags | Questions | StackExchange Sites | 355 | 1.233449 | 0.121528 |
| sha | Characters | Stage Groups | Shakespeare's Plays | 37 | 0.554054 | 0.304688 |
| syn-c | Hypergraph Configuration Models | | | 250 | 0.5 | 0.005 |
| syn-r | Erdős-Rényi Random Hypergraph Models | | | 250 | 0.5 | 0.005 |
| syn-s | Hypergaph Stochastic Block Models | | | 250 | 0.5 | 0.005 |

## 4  EXPERIMENTS

Having established in Section 3 that ORCHID curvatures have our desired theoretical properties, and finding that they strictly generalize both ORC on graphs and existing definitions of hypergraph ORC, we now seek to ascertain that they are also meaningful in practice. We ask the following questions:

**Q1 Parametrization.** How do our choices of $\alpha$, $\mu$, and AGG impact ORCHID curvatures?

**Q2 Hypergraph exploration.** How can ORCHID curvatures help us in exploring hypergraphs?

**Q3 Hypergraph learning.** How can ORCHID curvatures help us in hypergraph learning tasks?

To address these questions, we experiment with data from different domains, spanning several orders of magnitude. We investigate four *individual real-world hypergraphs* in which edges represent co-authorship (aps-a, dblp) and FDA-registered drugs (ndc-ai, ndc-pc), six *collections of real-world hypergraphs* in which edges represent questions on Stack Exchange Sites (stex), co-authorship by venues (aps-av, dblp-v), co-citation by venues (aps-cv), chords in music pieces (mus), and character cooccurrence on stage in Shakespeare's plays (sha), as well as three *collections of synthetic hypergraphs* based on different generative models (syn-c, syn-r, syn-s), for a total of 4 321 hypergraphs. We summarize their basic properties in Table 1, and give more details on their statistics, semantics, and provenance in Appendix A.3. We implement ORCHID in Julia and Python. Our experiments are run on AMD EPYC 7702 CPUs with up to 256 cores. We discuss our implementation and results in more detail in Appendices A.4 and A.5, and make all our code, data, and results publicly available.[1]

**Q1 Parametrization.**  To understand how our choices of $\alpha$, $\mu$, and AGG impact ORCHID curvatures, we first compute the pairwise mutual information between ORCHID edge curvatures with 36 different parametrizations. As illustrated in Fig. 2, while changing $\alpha$ for the same combination of $\mu$ and AGG has similar effects across hypergraphs, there is no uniform pattern in the relationships between different combinations of $\mu$ and AGG. This underscores the fact that the various notions of ORCHID curvature are not redundant but rather emphasize distinct aspects of hypergraph structure. For a fine-grained view of the differences between parametrizations, we inspect the distributions of

---

[1]https://doi.org/10.5281/zenodo.7624573

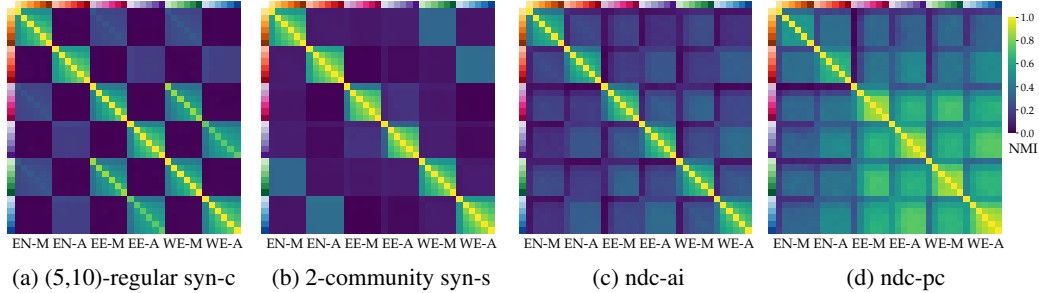

| (a) (5,10)-regular syn-c | (b) 2-community syn-s | (c) ndc-ai | (d) ndc-pc |

Figure 2: ORCHID curvature notions are non-redundant. We show the Min-Max-Normalized Mutual Information (NMI) between ORCHID edge curvatures with 36 different parametrizations, using probability measures $\mu^{\text{EN}}$ (EN), $\mu^{\text{EE}}$ (EE), or $\mu^{\text{WE}}$ (WE), aggregations $\text{AGG}_{\text{M}}$ (M) or $\text{AGG}_{\text{A}}$ (A), and $\alpha \in \{0.0, 0.1, 0.2, 0.3, 0.4, 0.5\}$ (ordered $\rightarrow, \downarrow$), for two synthetic and two real-world hypergraphs.

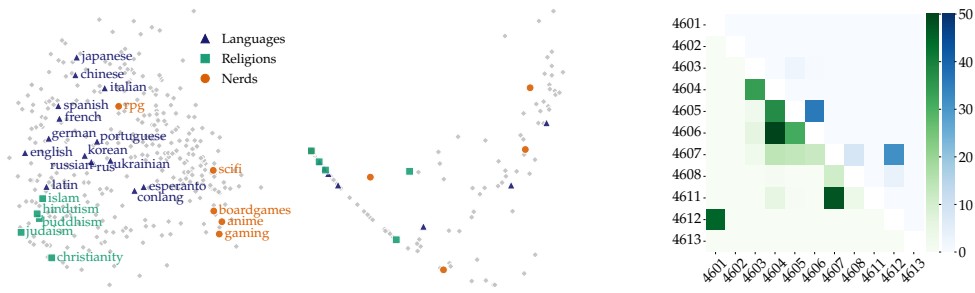

| (a) kPCA (directional curvature) | (b) kPCA (edge neighborhood size) | (c) MMD (cardinality vs. curvature) |

Figure 3: Curvatures carry more information than other local features. We show a 2-dimensional embedding of graphs from the stex collection based on kPCA, using an RBF kernel with curvature distributions computed using $\alpha = 0.1$, $\mu^{\text{WE}}$, and $\text{AGG}_{\text{A}}$ (3a) or edge neighborhood size distributions (3b) as input features. We see that only curvatures yield a meaningful and discriminative grouping. Corroborating this finding, we also depict Bonferroni-adjusted p-values of testing for significant differences in feature distributions—i.e., p-values multiplied by the number $h$ of hypothesis tests, as Bonferroni (1936) correction requires $p \leq \alpha/h$ for some desired Type I-error rate $\alpha$—using MMD on distributions of edge curvatures computed with the same parameters as for (3a) (upper triangle) or edge cardinality (lower triangle), for the subset of the dblp-v collection corresponding to top conferences grouped by areas of research (3c).

our four curvature types, (i) edge curvature $\kappa(e)$, (ii) edge-averaged node curvature $\kappa^{E}(i)$, (iii) directional curvature $\kappa(i, j)$ for all $\{i, j\} \subseteq e \in E$, and (iv) direction-averaged node curvature $\kappa^{\mathcal{N}}(i)$, for each of our 36 parametrizations. By construction, directional curvature and direction-averaged node curvature do not vary with the choice of AGG, and $\kappa_{\text{M}}$ lower-bounds $\kappa_{\text{A}}$ for edge curvatures and edge-averaged node curvatures. However, the differences between $\kappa_{\text{M}}$ and $\kappa_{\text{A}}$ vary across graphs, while consistently, the larger $\alpha$, the more concentrated our curvature distributions (Appendix A.5).

**Q2 Hypergraph Exploration.** To explore *individual graphs*, we perform case studies on graphs from the aps-cv collection, leveraging that most nodes in these graphs also occur as edges. We scrutinize the relationships between node and edge curvatures, other local node and edge statistics, and article metadata. We observe that curvature values span a considerable range even for articles with otherwise comparable statistics, but the curvature distributions of influential papers appear to differ systematically from those of less influential papers (Appendix A.5). Exploring *graph collections*, we run kernel PCA (kPCA) (Schölkopf et al., 1997) with a radial basis function kernel (RBF kernel) and curvatures or other local features known to be powerful baselines (Cai & Wang, 2018), e.g., node degrees and neighborhood sizes, as inputs to jointly embed graphs from a collection. We statistically bootstrap the maximum mean discrepancy (MMD) (Gretton et al., 2006) to test the null hypothesis that the feature distributions of two graphs are equal. As shown in Fig. 3, ORCHID curvatures result in more interpretable embeddings and more discriminative tests than other local features.

Table 2: ORCHID curvatures lead to better clusterings than other local features. We show $\text{WCC}_{\kappa(i,j)}$ for collection clusterings computed using RBF or exp. Wasserstein kernels with edge curvatures, edge neighborhood sizes, edge-averaged node curvatures, or node neighborhood sizes as inputs.

|        | $\text{RBF}_{\kappa(e)}$ | $\text{W}_{\kappa(e)}$ | $\text{RBF}_{\|\mathcal{N}(e)\|}$ | $\text{W}_{\|\mathcal{N}(e)\|}$ | $\text{RBF}_{\kappa^E(i)}$ | $\text{W}_{\kappa^E(i)}$ | $\text{RBF}_{\|\mathcal{N}(i)\|}$ | $\text{W}_{\|\mathcal{N}(i)\|}$ |
|--------|--------|--------|--------|--------|--------|--------|--------|--------|
| dblp-v | 0.2151 | 0.1908 | 0.3309 | 0.2358 | 0.2273 | **0.0445** | 0.0910 | 0.1285 |
| mus    | 0.1955 | 0.1758 | 0.2609 | 0.2723 | 0.2062 | **0.1606** | 0.2774 | 0.2458 |
| stex   | 0.2651 | 0.2877 | 0.3018 | 0.2950 | **0.2393** | 0.2577 | 0.3067 | 0.2689 |
| sha    | 0.5984 | 0.6390 | 0.6716 | 0.6597 | **0.5021** | 0.6526 | 0.6236 | 0.6641 |

**Q3 Hypergraph Learning.** To explore the utility of curvatures for learning on *individual hypergraphs*, we perform spectral clustering using either curvatures or other local node features. To evaluate the resulting node clusterings, we leverage that *nodes* in the aps-cv collection correspond to APS papers, for which we consistently know the titles. Hence, even in the absence of a meaningful ground truth, we can still check the sensibility of a clustering by statistically analyzing the titles grouped together using tools from natural language processing. We find that node clusterings based on curvatures correspond to thematically more coherent groupings (Appendix A.5). For learning on *hypergraph collections*, we spectrally cluster the collection using RBF or exponential Wasserstein kernel matrices, $\exp(-\gamma\, \text{W}(\mu_x, \mu_y))$, on node and edge curvatures or other local features (Plaen et al., 2020). Lacking ground-truth labels, we evaluate the clustering quality in an *unsupervised* manner, using what we call the *Wasserstein Clustering Coefficient* (WCC). This measure compares averaged *intra*-cluster Wasserstein distances to averaged *inter*-cluster Wasserstein distances, such that a *lower* WCC corresponds to a higher-quality clustering. Given $c$ clusters $\mathcal{X} = \{X_1, \ldots, X_c\}$ of hypergraphs $H$ represented by their feature distributions $\vec{\chi}_H$, we define

$$\text{WCC}(\mathcal{X}) := \frac{\sum_{X \in \mathcal{X}} \omega(X)}{1 + \sum_{X \neq Y \in \mathcal{X}} \omega(X, Y)}\ , \ \text{with} \begin{cases} \omega(X) := \binom{|X|}{2}^{-1} \sum_{x \neq y \in X} \text{W}(\vec{\chi}_x, \vec{\chi}_y)\ , \\ \omega(X, Y) := (|X||Y|)^{-1} \sum_{x,y \in X \times Y} \text{W}(\vec{\chi}_x, \vec{\chi}_y)\ . \end{cases}$$

As illustrated in Table 2, when evaluated using WCC with directional curvature distributions as $\vec{\chi}$, i.e., $\text{WCC}_{\kappa(i,j)}$, ORCHID curvatures consistently yield better clusterings than other local features.

## 5 DISCUSSION AND CONCLUSION

We introduced ORCHID, the first unified framework for Ollivier-Ricci curvature on hypergraphs that integrates and generalizes existing approaches to hypergraph ORC. ORCHID disentangles the common building blocks of all notions of hypergraph ORC, yielding curvature notions that are provably aligned with our geometric intuition. We performed a rigorous theoretical and empirical analysis of ORCHID curvatures, demonstrating their practical utility and scalability through extensive experiments, covering both *hypergraph exploration* and *hypergraph learning*. While our work paves the way toward future work seeking to leverage the power of Ollivier-Ricci curvature for hypergraphs in hypergraph learning algorithms, it still has some limitations to be addressed. First, ORC on graphs is defined for *any* probability measure, but we only consider measures corresponding to a single step of a random walk. Future work could thus harness higher-order random walks or alternative probability measures, and consider analyzing relationships between such probability measures and other structural hypergraph properties. Second, hyperedge intersections can vary in cardinality, but this variation is not currently reflected in our probability measures. One could thus integrate ORCHID with the $s$-walk framework proposed by Aksoy et al. (2020), or define persistent ORCHID curvatures based on hypergraph filtrations, extending work on persistent ORC for graphs (Wee & Xia, 2021b). Third, like the original ORC, ORCHID curvatures are static, but many hypergraphs are inherently dynamic, suggesting a need to develop dynamic curvature notions. Fourth, despite its comprehensive scope, our study only scratches the surface regarding the theoretical and empirical analysis of ORCHID curvatures, and we believe that there are many more connections between ORCHID curvatures and other hypergraph descriptors to be uncovered, and many additional use cases to be explored. For instance, ORCHID generalizes ORC, but not Forman–Ricci curvature (FRC), and we believe that a framework for FRC could help uncover new relations between combinatorial curvature notions and hypergraph structure. Finally, we imagine that incorporating hypergraph curvature into models as an additional inductive bias could prove useful in hypergraph learning more broadly.

## ETHICS STATEMENT

Our main contribution is ORCHID, a unified mathematical framework yielding theoretically sound hypergraph descriptors that are also practically useful for hypergraph exploration and hypergraph learning. As such, ORCHID comes with the caveats applicable to hypergraph exploration and hypergraph learning methods more generally. Most importantly, it should be used with caution on data related to people, and its results should not be decontextualized. We adhered to these principles in our experiments, and selected our datasets accordingly.

## REPRODUCIBILITY STATEMENT

To facilitate reproducibility, we provide more details on our data, implementation, and results in more detail in Appendices A.3 to A.5, and make all our code, data, and results publicly available at https://doi.org/10.5281/zenodo.7624573.

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

## A  APPENDIX

In this Appendix, we include the following materials.

**A.1  Deferred Proofs.**
All proofs for Section 3, along with supporting definitions, lemmas and corollaries.

**A.2  Related Work.**
Discussion of related work treating hypergraph curvatures, graph curvatures, or hypergraph analysis.

**A.3  Dataset Details.**
Further information on the provenance, semantics, and statistics of our datasets.

**A.4  Implementation Details.**
Details on our implementation, including proofs showing the correctness of performance shortcuts.

**A.5  Further Results.**
Display and discussion of results not included in the main paper.

### A.1  DEFERRED PROOFS

**Lemma 1.** *For graphs and $r$-uniform, $k$-regular, $c$-cooccurrent hypergraphs, $\mu^{\mathrm{EN}} = \mu^{\mathrm{EE}} = \mu^{\mathrm{WE}}$.*

*Proof.* For notational simplicity, w.l.o.g., we assume that $\alpha = 0$. In an $r$-uniform, $k$-regular, $c$-cooccurrent hypergraph $H = (V, E)$, each node $i$ has degree $k$ and $\frac{(r-1)k}{c}$ neighbors, and each edge has cardinality $r$. Hence, for nodes $i, j \in V$ with $i \sim j$,

$$\mu_i^{\mathrm{EN}}(j) = \frac{1}{|\mathcal{N}(i)|} = \frac{c}{(r-1)k} = \frac{1}{k} \cdot c \cdot \frac{1}{r-1} = \frac{1}{\deg(i)} \sum_{e \ni i,j} \frac{1}{|e|-1} = \mu_i^{\mathrm{EE}}(j)$$

$$= \frac{c}{k(r-1)} = \frac{|\{e \in E \mid \{i,j\} \subseteq e\}|}{\sum_{f \ni i} (|f|-1)} = \mu_i^{\mathrm{WE}}(j) \,.$$

Graphs are 2-uniform and 1-cooccurrent (but not generally regular), and hence, $|\mathcal{N}(i)| = \deg(i)$. Using this to simplify the probability measure expressions, the claim follows. $\square$

**Lemma 2.** *For graphs, i.e., 2-uniform hypergraphs, we have $\mathrm{AGG_A}(e) = \mathrm{AGG_B}(e) = \mathrm{AGG_M}(e)$ for all edges $e \in E$.*

*Proof.* Given probability distributions $\mu_1, \mu_2, \ldots, \mu_n$, their Wasserstein barycenter is defined as the distribution $\bar{\mu}$ that minimizes $f(\bar{\mu}) := \frac{1}{n} \sum_{i=1}^{n} \mathrm{W}_1(\bar{\mu}, \mu_i)$. Since $|e| = 2$, we minimize $\mathrm{W}_1(\bar{\mu}, \mu_1) + \mathrm{W}_1(\bar{\mu}, \mu_2)$. The Wasserstein distance is a metric, so it satisfies the triangle inequality. Thus, $\mathrm{W}_1(\mu_1, \mu_2) \leq \mathrm{W}_1(\bar{\mu}, \mu_1) + \mathrm{W}_1(\bar{\mu}, \mu_2)$ for all choices of $\bar{\mu}$. Hence, $f$ is minimized by either $\mu_1$ or $\mu_2$. Evaluating both cases yields $\mathrm{AGG_A}(e) = \mathrm{AGG_B}(e)$, and observing that $\mathrm{AGG_M}(e) = \mathrm{W}_1(\mu_i, \mu_j)$ for $e = \{i, j\}$ by definition, the claim follows. $\square$

**Theorem 3.** *For any probability measure $\mu$ and $C(e) := {}^2/_{|e|(|e|-1)}$, the curvature $\kappa_{\mathrm{A}}(e)$ of an edge $e \in E$ is bounded by*

$$1 - \mathrm{diam}(H) C(e) \sum_{\{i,j\} \subseteq e} \|\mu_i - \mu_j\|_1 \leq \kappa_{\mathrm{A}}(e) \leq 1 - \mathrm{d_{min}}(H) C(e) \sum_{\{i,j\} \subseteq e} \|\mu_i - \mu_j\|_1 \,. \tag{15}$$

*Proof.* We bound each of the summands in the curvature calculation. Given probability measures $\mu_i, \mu_j$, a result by Gibbs & Su (2002, Theorem 4) states that

$$\mathrm{d_{min}}(H) \, \mathrm{d_{TV}}(\mu_i, \mu_j) \leq \mathrm{W}_1(\mu_i, \mu_j) \leq \mathrm{diam}(H) \, \mathrm{d_{TV}}(\mu_i, \mu_j) \,, \tag{18}$$

where $\mathrm{d_{TV}}$ refers to the *total variation distance*. The intuition behind this bound is that the total variation distance represents a specific type of transport plan between the two probability measures; the factors arising from the minimum (maximum) distance in a space indicate the minimum (maximum) distance that realizes this transport plan. Since all our measures are defined over a finite space, we have $\mathrm{d_{TV}}(\mu_i, \mu_j) = {}^1/_2 \|\mu_i - \mu_j\|_1$. The claim follows by considering that pairwise distances are being *subtracted* to calculate our curvature measure. $\square$

**Theorem 4.** *For any probability measure $\mu$, the curvature $\kappa_{\mathrm{M}}(e)$ of an edge $e \in E$ is bounded by*

$$1 - \mathrm{diam}(H) \max_{\{i,j\} \subseteq e} \|\mu_i - \mu_j\|_1 \leq \kappa_{\mathrm{M}}(e) \leq 1 - \mathrm{d}_{\min}(H) \max_{\{i,j\} \subseteq e} \|\mu_i - \mu_j\|_1 . \qquad (16)$$

*Proof.* For $\textsc{Agg}_{\mathrm{M}}$, Eq. (18) applies for a single pairwise distance only. We thus only obtain a single bound based on the maximum total variation distance between two probability measures. $\qquad\square$

**Theorem 7.** *Given a subset of nodes $s \subseteq V$ and an arbitrary probability measure $\mu$, let $\delta_i$ denote a Dirac measure at node $i$, and let $\mathrm{J}(\mu_i) := \mathrm{W}_1(\delta_i, \mu_i)$ denote the* jump probability *of $\mu_i$. If (i) all curvatures based on $\mu$ are strictly positive, i.e., $\kappa(s) \geq \kappa > 0$ for all $s \subseteq V$, and (ii) $\mathrm{W}_1(\mu_i, \mu_j) \leq \textsc{Agg}(s)$ for $\{i, j\} = \mathrm{argmax}(\mathrm{d}(s))$, then*

$$\mathrm{d}(s) \leq \frac{\mathrm{J}(i) + \mathrm{J}(j)}{\kappa(s)} . \qquad (17)$$

*Proof.* Let $\{i, j\} = \mathrm{argmax}(\mathrm{d}(s))$ as required in the theorem. We then have following chain of (in)equalities:

$$\mathrm{d}(s) = \mathrm{d}(i, j) = \mathrm{W}_1(\delta_i, \delta_j) \leq \mathrm{W}_1(\delta_i, \mu_i) + \mathrm{W}_1(\mu_i, \mu_j) + \mathrm{W}_1(\mu_j, \delta_j) . \qquad (19)$$

Rearranging Eq. (14), we have $(1 - \kappa(s))\, \mathrm{d}(s) = \textsc{Agg}(s)$. According to our assumptions, $\mathrm{W}_1(\mu_i, \mu_j) \leq \textsc{Agg}(s) = (1 - \kappa(s))\, \mathrm{d}(i, j)$. Inserting this into Eq. (19) yields

$$\mathrm{d}(i, j) \leq \mathrm{J}(\mu_i) + \mathrm{J}(\mu_j) + (1 - \kappa(s))\, \mathrm{d}(i, j) \qquad (20)$$
$$\Leftrightarrow \qquad \mathrm{d}(i, j) - (1 - \kappa(s))\, \mathrm{d}(i, j) \leq \mathrm{J}(\mu_i) + \mathrm{J}(\mu_j) \qquad (21)$$
$$\Leftrightarrow \qquad \mathrm{d}(i, j) \leq \frac{\mathrm{J}(i) + \mathrm{J}(j)}{\kappa(s)} , \qquad (22)$$

where the last step is only valid since $\kappa(s) \geq \kappa > 0$ by assumption. $\qquad\square$

**Definition 11** (Hypercliques, hypergrids, hypertrees). *A simple, connected hypergraph $H = (V, E)$ is*

- *a* hyperclique *if $E = \binom{V}{r}$ for some $r \leq |V|$,*

- *a* hypergrid *if $H$ is an $r$-uniform hypergraph for which there exists a lattice $L = (V, E_L)$ s.t. $E = \{e \in \binom{V}{r} \mid e$ corresponds to a path of length $r$ in $L\}$, and*

- *a* hypertree *if there exists a tree $T = (V, E_T)$ s.t. each edge $e \in E_T$ induces a subtree in $T$.*

**Corollary 12.** *Cliques are hypercliques, grids are hypergrids, and trees are hypertrees.*

**Corollary 13.** *If $H = (V, E)$ is a hyperclique, a hypergrid, or an $r$-uniform, $k$-regular, 1-intersecting hypertree, for $i, j \in V$, the sets $S_i = \{e \in E \mid i \in e\}$ and $S_j = \{e \in E \mid j \in e\}$ are isomorphic, i.e., there exists $\varphi : \mathcal{N}(i) \cup \{i\} \to \mathcal{N}(j) \cup \{j\}$ such that $\{\{\varphi(x) \mid x \in e\} \mid e \in S_i\} = S_j$.*

For hypercliques, hypergrids, and hypertrees with certain regularities, $\textsc{Agg}_{\mathrm{A}}(e)$ and $\textsc{Agg}_{\mathrm{M}}(e)$ are constants.

**Lemma 14** (Hypercliques, hypergrids, hypertrees). *If $H = (V, E)$ is a hyperclique, a hypergrid, or an $r$-uniform, $k$-regular, 1-intersecting hypertree, we have $\textsc{Agg}_{\mathrm{A}}(e) = \textsc{Agg}_{\mathrm{M}}(e) = \mathrm{W}_1(\mu_i, \mu_j) = w$ for $w \in \mathbb{R}$, $e \in E$, and $i, j \in V$ with $i \sim j$.*

*Proof.* By Corollary 13, we have $w := \mathrm{W}_1(\mu_i, \mu_j) = \mathrm{W}_1(\mu_p, \mu_q)$ for $i, j, p, q \in V$ with $i \sim j$ and $p \sim q$. Hence $\textsc{Agg}_{\mathrm{M}}(e) = w$, and $\textsc{Agg}_{\mathrm{A}}(e) = \frac{2}{|e|(|e|-1)} \sum_{\{i,j\} \subseteq e} \mathrm{W}_1(\mu_i, \mu_j) = \frac{2}{|e|(|e|-1)} \frac{|e|(|e|-1)}{2} w = w$, for $e \in E$. $\qquad\square$

**Corollary 15.** *If $H = (V, E)$ is a hyperclique, a hypergrid, or an $r$-uniform, $k$-regular, 1-intersecting hypertree, $\textsc{Agg}_{\mathrm{A}}(e) = \textsc{Agg}_{\mathrm{M}}(e)$.*

Using Lemma 14, we now prove that under $\text{AGG}_\text{A}$ and $\text{AGG}_\text{M}$, hypercliques are positively curved, hypergrids are flat, and hypertrees are negatively curved, as desired.

**Theorem 8** (Hyperclique curvature). *For an edge $e$ in a hyperclique $H = (V, E)$ on $n$ nodes with edges $E = \binom{V}{r}$ for some $r \leq n$, with $\alpha = 0$,*

$$\kappa(e) = 1 - \frac{1}{n-1}, \; i.e., \; \lim_{n \to \infty} \kappa(e) = 1, \; independent \; of \; r.$$

*Proof.* A hyperclique is $r$-uniform, $(n-1)$-regular, and $(r-2)$-cooccurrent, so $\mu_i^{\text{EN}} = \mu_i^{\text{EE}} = \mu_i^{\text{WE}}$ for each node $i \in V$ by Lemma 1. Thus, considering $\mu_i^{\text{EN}}$, each node $i \in V$ has $n - 1$ neighbors to which it distributes its probability mass equally, and we have $W_1(\mu_i, \mu_j) = \frac{1}{n-1}$ for $i, j \in V$ with $i \sim j$. The claim now follows from Lemma 14. $\square$

**Theorem 9** (Hypergrid curvature). *For an edge $e$ in a $r$-uniform, $k$-regular hypergrid, with $\alpha = 0$, $\kappa(e) = 0$, independent of $r$ and $k$.*

*Proof.* By Corollary 13, the sets $S_i = \{e \in E \mid i \in e\}$ and $S_j = \{e \in E \mid j \in e\}$ are isomorphic, and due to the symmetries in the hypergrid, the isomorphism $\varphi \colon \mathcal{N}(i) \cup \{i\} \to \mathcal{N}(j) \cup \{j\}$ minimizing the cost $\sum_{x \in \mathcal{N}(i) \cup \{i\}} \mathrm{d}\left(x, \varphi(x)\right)$ corresponds to the coupling minimizing $W_1(\mu_i, \mu_j)$. The cost of $\varphi$ equals the minimum cost of an isomorphism in $H$'s underlying lattice $L$ between the inclusive $(r-1)$-hop neighborhoods of two nodes adjacent in $L$, which is $|\mathcal{N}(i) \cup \{i\}|$. Hence, $W_1(\mu_i, \mu_j) = \frac{|\mathcal{N}(i) \cup \{i\}|}{|\mathcal{N}(i) \cup \{i\}|} = 1$ for $i, j \in V$ with $i \sim j$ and all choices of $\mu$, and the claim then follows from Lemma 14. $\square$

**Theorem 10** (Hypertree curvature). *For an edge $e$ in a $r$-uniform, $k$-regular, $1$-intersecting hypertree,*

$$with \; \alpha = 0, \; \kappa(e) = 1 - \left(\frac{3(k-1)}{k} + \frac{1}{(r-1)k}\right), \; i.e., \; \lim_{k \to \infty} \kappa(e) = -2, \; independent \; of \; r.$$

*Proof.* An $r$-uniform, $k$-regular, $1$-intersecting hypertree is $1$-cooccurrent, so we have $\mu_i^{\text{EN}} = \mu_i^{\text{EE}} = \mu_i^{\text{WE}}$ for each node $i \in V$ by Lemma 1. Each node $i \in V$ has $(r-1)k$ neighbors, such that $\mu_i^{\text{EN}}$ distributes a fraction $\frac{1}{(r-1)k}$ of the probability mass to each of $i$'s neighbors. Nodes $i, j \in V$ with $i \sim j$ share $(r-2)$ neighbors (those in the unique edge $e$ satisfying $\{i, j\} \subseteq e$), and the probability mass allocated by $\mu_i$ to $j$ can be matched with the probability mass allocated by $\mu_j$ to $i$ at cost 1. Because $H$ is a hypertree, the remaining probability mass, $(r-1)(k-1)/\left((r-1)k\right) = (k-1)/k$, needs to be transported from the neighborhood of $i$ to the neighborhood of $j$ at cost 3. Hence,

$$W_1(\mu_i, \mu_j) = 1 \cdot \frac{1}{(r-1)k} + 3 \cdot \frac{k-1}{k}$$

for $i, j \in V$ with $i \sim j$. Again, the claim follows from Lemma 14. $\square$

## A.2 RELATED WORK

**Hypergraph Curvature**   Most closely related to our work is the literature on hypergraph curvatures. Much of this literature focuses on defining notions of ORC and Forman-Ricci Curvature (FRC) specifically for *directed* hypergraphs and studying some of their mathematical and empirical properties (e.g., Leal et al., 2019; 2020; 2021; Saucan & Weber, 2018). Notably, the directed hypergraph ORC introduced by Eidi & Jost (2020) is an instantiation of our framework with $\mu^{\text{EE}}$ and $\texttt{AGG}_{\texttt{A}}$. Curvature notions for *undirected* hypergraphs are comparatively less explored, and especially the literature generalizing ORC is almost entirely theoretical. The generalization of ORC proposed by Asoodeh et al. (2018) and the equivalent measure used by Banerjee (2021) are instantiations of our framework using $\mu^{\text{EE}}$ and $\texttt{AGG}_{\texttt{B}}$. Akamatsu (2022) propose $(\alpha, h)$-ORC using cost functions based on structured optimal transport, and Ikeda et al. (2021) define $\lambda$-coarse Ricci curvature using a $\lambda$-nonlinear Kantorovich difference based on a submodular hypergraph Laplacian as a generalization of ORC as introduced by Lin et al. (2011). Both of these works define curvature exclusively for pairs of nodes, rather than for hyperedges. Beyond ORC, Yadav et al. (2022) study FRC for undirected hypergraphs defined via poset representations, and Murgas et al. (2022) explore hypergraphs constructed from protein-protein interactions using a different notion of FRC based on the Hodge Laplacian. To the best of our knowledge, with ORCHID, we are the first to introduce a flexible framework generalizing ORC to hypergraphs, and to demonstrate the utility of hypergraph ORC in practice.

**Graph Curvature.**   Beyond the Ollivier-Ricci concepts, there are also curvature concepts based on the contractivity of operators (Bakry & Émery, 1985), which could be considered a "spiritual precursor" to Ollivier's work. This perspective has been used to provide a predominantly *spectral perspective* on curvature (Liu et al., 2019; Münch & Rose, 2020), whereas ORC can foremost be seen as a *probabilistic concept*. Recently, Kempton et al. (2020) defined a hybrid between Ollivier and Bakry-Émery curvature on graphs. A more combinatorial perspective is assumed by FRC, which is motivated by defining equivalent formulations of curvature on structured spaces, such as CW complexes or simplicial complexes. Originally described by Forman (2003), FRC has since been improved in the context of explaining the learning behavior of graph neural networks (Topping et al., 2022), with other recent work focusing on fusing it with topological graph properties (Roy et al., 2020). ORC was first developed for general Markov chains (Ollivier, 2007; 2009), but has quickly been adopted to characterize graphs (Jost & Liu, 2014) and networks (Weber et al., 2017). With numerous follow-up publications elucidating the relationship between structural properties of a graph and ORC (Bauer et al., 2017; Samal et al., 2018), the initial concept has also been substantially updated (Bourne et al., 2018; Lin et al., 2011). As an emerging research direction, we identified the combination of ORC (and FRC) with concepts from computational topology, leading to an inherent multi-scale perspective on data. This has led to promising results for treating biomedical graph data (Wee & Xia, 2021a;b).

**Hypergraph Learning.**   Work tackling certain hypergraph learning tasks such as hypergraph clustering (Amburg et al., 2020; Veldt et al., 2020) has existed for many years (Wachman & Khardon, 2007; Zhou et al., 2006). Some approaches make use of intrinsic structural properties of hypergraphs, leading to hypergraph neural network architectures (Huang & Yang, 2021) and message passing formulations (Gao et al., 2019), whereas others focus on developing similarity measures, i.e., *kernels* (Bai et al., 2014; Bloch & Bretto, 2013; Martino & Rizzi, 2020). Methods from the rich literature on *graph* kernels can also be employed to address hypergraph learning tasks, namely, by transforming the hypergraph into a graph, but most popular transformations are lossy and may drastically increase the size of the object under study, such that the practicality and utility of this approach is unclear.

**Hypergraph Mining and Analysis.**   In recent years, there has been a renewed interest in hypergraph mining and analysis. Notably, there is work developing new hypergraph descriptors (Aksoy et al., 2020), extending motif discovery to hypergraphs (Lee & Shin, 2021; Lee et al., 2020), solving classic graph mining tasks in the hypergraph setting (Macgregor & Sun, 2021), or identifying patterns in real-world hypergraphs (Do et al., 2020). However, to the best of our knowledge, none of this work draws on curvature concepts to solve the mining and analysis tasks of interest.

### A.3 Dataset Details

At a high level, our workflow to produce and work with the datasets used in our experiments (Section 4) was as follows:

1. Obtain raw data in a variety of different formats, e.g., CSV, JSON, or XML.

2. Transform the raw data into a hypergraph CSV that retains as much of the raw data semantics as possible. This CSV is guaranteed to contain one row per edge, one column with unique edge identifiers, and one column with the nodes contained in each edge. It may also contain additional columns holding further metadata associated with individual edges. Column names may differ between datasets to reflect dataset semantics.

3. Provide a unified loading interface to the datasets in Python.

4. Transform semantics-laden hypergraph CSV files into semantics-free one-based integer edge lists and sparse matrices for curvature computations in Julia, compute curvatures in Julia, and store the results in JSON files.

5. Map results back to original dataset semantics in Python for further examination.

In the following, we give more details on the provenance, semantics, and statistics of our datasets. Unless if otherwise noted, we make our datasets publicly available with our online materials, along with the raw data and all preprocessing code.[2]

#### A.3.1 APS-A, APS-AV, APS-CV: American Physical Society Journal Articles

The American Physical Society (APS), a nonprofit organization working to advance the knowledge of physics, publishes several peer-reviewed research journals. The APS makes two datasets based on its publications available to researchers: (i) an edge list containing (citing, cited) pairs of articles contained in its collection, and (ii) a JSON dataset containing the metadata for each article in its collection. These datasets are updated on a yearly basis, and researchers can request access by filling out a web form located at `https://journals.aps.org/datasets`. We made a data access request and were granted access to the 2021 versions of the APS datasets within two weeks.

From the APS datasets, we derived the following hypergraphs and hypergraph collections:

(i) aps-a: Each node corresponds to an author who published at least one article in an APS journal. Each edge $e$ corresponds to an article in an APS journal, and it contains as nodes all authors of $e$. This hypergraph is derived from the JSON data.

(ii) aps-av: aps-a, split up by journal, for a total of 19 hypergraphs. For each journal $j$, the edge set of aps-a is restricted to articles from $j$, and the node set of aps-a is restricted to nodes authoring at least one article from $j$.

(iii) aps-cv: We derive one hypergraph for each of the 19 journals represented in the edge list data. For each journal $j$, the edge set comprises articles from $j$ citing at least one article in $j$, and the node set consists of articles in $j$ cited by at least one article in $j$.

**Access.**  Due to the terms and conditions associated with data access, we cannot make the APS datasets or the hypergraphs derived from them publicly available, and researchers seeking to work with this data will have to request data access from APS directly as outlined above. However, we make our preprocessing code publicly available, such that researchers who have obtained access to the APS datasets can easily reproduce our hypergraphs from the raw data.

**Caveats.**  When doing our case studies on the aps-cv dataset, we observed that some DOIs present in the edge list had no associated metadata in the JSON files provided by APS. This does not affect our curvature computations, but it might constrain the interpretability of results, e.g., when inspecting node clustering results based on article categories present only in the metadata.

---

[2]https://doi.org/10.5281/zenodo.7624573

A.3.2  DBLP, DBLP-V: DBLP JOURNAL ARTICLES AND CONFERENCE PROCEEDINGS

The DBLP computer science library provides high-quality bibliographic information on computer science publications. All DBLP data is released under a CC0 license and freely available in one XML file that is updated regularly. We obtained the XML dump dated September 1, 2022 from `https://dblp.org/xml/release/` and preprocessed it into a CSV file containing only entries corresponding to the XML tags `article` and `inproceedings`, with one row per entry and the following columns:

- key: unique identifier of the entry, e.g., `conf/iclr/XuHLJ19` or `journals/cacm/Savage16c`.
- tag: XML tag associated with the entry, one of {`inproceedings`, `article`}.
- crossref: cross-reference to a venue, e.g., `conf/iclr/2019`. Sometimes missing although a venue should be present.
- author: semicolon-separated list of DBLP author names, e.g., `Keyulu Xu;Weihua Hu;Jure Leskovec;Stefanie Jegelka`. Sometimes missing (we discard entries without authors when loading the data).
- year: entry publication year, e.g., `2019`.
- title: entry title, e.g., `How Powerful are Graph Neural Networks?`.
- publtype: if present, the type of publication, e.g., `informal`. Mostly missing.
- journal: for article entries, the name of the publishing journal, e.g., `Commun. ACM`.
- booktitle: for inproceedings entries, the name of the publishing venue, e.g., `ICLR`.
- volume: if present, the publication volume, e.g., `59`.
- number: if present, the publication number, e.g., `7`.
- pages: if present, the entry pages, e.g., `12-14`.
- mdate: modification date, e.g., `2019-07-25`.

This constitutes our individual hypergraph dblp, in which each edge represents a paper, and each node represents an author. From this hypergraph, we additionally derived the dblp-v hypergraph collection, which contains different subsets of dblp by venue or group of venues. More precisely, we distinguish 1 193 hypergraphs as follows:

(i) `dblp_journal-all`, `dblp_inproceedings-all`: partition of dblp into entries published in journals and entries published as part of proceedings.

(ii) `dblp_journal-{journal}`: one hypergraph per journal, for all journals with at least 1 000 articles in the DBLP dataset.

(iii) `dblp_proceedings-{venue}`: one hypergraph per venue (grouped by `booktitle`), for all venues with at least 1 000 papers in the DBLP dataset.

(iv) `dblp_proceedings_area-{area}_{venues}`: one hypergraph per each of the FoR (field of research) areas 4601–4608, 4611–4613 as used in the CORE ranking (4609 and 4610 were not present in the ranking), where each area is represented by all conferences (grouped by `booktitle`) with CORE rank A* and A that have at least 1 000 papers in the DBLP dataset. These areas and associated top conferences are as follows:

- 4601: Applied computing – AIED, ICCS
- 4602: Artificial intelligence – AAAI, AAMAS, ACL, AISTATS, CADE, CIKM, COLING, COLT, CP, CogSci, EACL, EC, ECAI, EMNLP, GECCO, ICAPS, IJCAI, IROS, KR, UAI
- 4603: Computer vision and multimedia computation – AAAI, CVPR, ECAI, ICCV, ICME, IJCAI, IROS, WACV
- 4604: Cybersecurity and privacy – AsiaCCS, CCS, CRYPTO, DSN
- 4605: Data management and data science – CIKM, ECIR, EDBT, ICDAR, ICDE, ICDM, ISWC, KDD, MSR, PODS, RecSys, SDM, SIGIR, VLDB, WSDM, WWW
- 4606: Distributed computing and systems software – ASPLOS, CCGRID, CLUSTER, CONCUR, DISC, DSN, HPCA, HPDC, ICCAD, ICDCS, ICNP, ICPP, ICS, ICWS, INFOCOM, IPDPS, IPSN, PODC, SC, SIGCOMM, SPAA, WWW
- 4607: Graphics, augmented reality and games – ISMAR, SIGGRAPH, VR, VRST

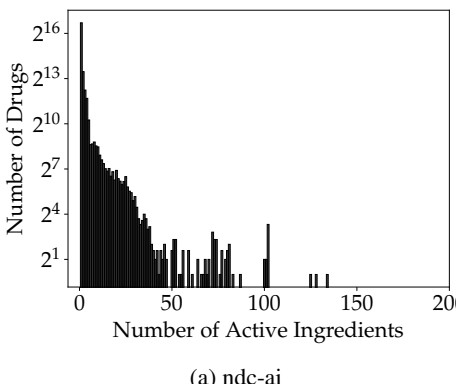

(a) ndc-ai

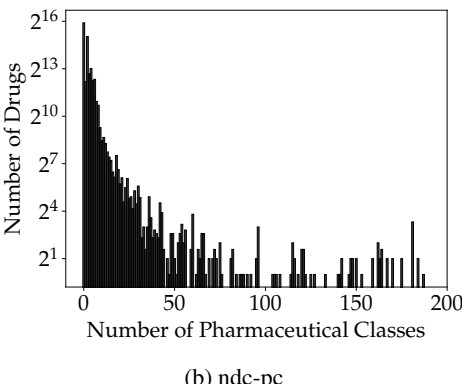

(b) ndc-pc

Figure 4: Edge cardinality distributions for hypergraphs derived from NDC data.

- 4608: Human-centred computing – ASSETS, CHI, CSCW, ITiCSE, IUI, SIGCSE, UIST
- 4611: Machine learning – AAAI, AISTATS, COLT, ECAI, ICDM, ICLR, ICML, IJCAI, KDD, NeurIPS, PPSN, WSDM
- 4612: Software engineering – ASE, ASPLOS, CAV, ICSE, ICST, ISCA, ISSRE, MSR, OOPSLA, PLDI, POPL, RE, SIGMETRICS
- 4613: Theory of computation – EC, ESA, FOCS, ICALP, ICLP, ISAAC, ISSAC, KR, LICS, MFCS, SODA, STACS, STOC, WG

**Caveats.** For about 0.1% of all records, our XML parser failed, which originally resulted in "None" as one of the authors of all problematic records. We then redid the preprocessing (and all subsequent computations) *excluding* those records, but the records were still counted when determining the venues to include in dblp-v.

### A.3.3 NDC-AI, NDC-PC: DRUGS APPROVED BY THE U.S. FOOD & DRUG ADMINISTRATION

The U.S. Food and Drug Administration (FDA) collects information on all drugs manufactured, prepared, propagated, compounded, or processed by registered drug establishments for commercial distribution in the United States. The FDA maintains the National Drug Code (NDC) Directory, which is updated daily and contains the listed NDC numbers and all information submitted as part of a drug listing. We downloaded the NDC data from `https://download.open.fda.gov/drug/ndc/drug-ndc-0001-of-0001.json.zip` on August 21, 2022, and transformed it into a CSV file, an example record of which is shown in Table 3. From this CSV file, we derived two hypergraphs. In both hypergraphs, edges correspond to FDA-registered drugs. In ndc-ai, nodes correspond to the active ingredients used in these drugs, and in ndc-pc, nodes correspond to the pharmaceutical classes assigned to these drugs. The edge cardinality distributions resulting from both semantics are shown in Fig. 4.

### A.3.4 MUS: MUSIC PIECES

`music21` is an open-source Python library for computer-aided musicology that comes with a corpus of public-domain music in symbolic notation. Using the `music21` library, we extracted a collection of hypergraphs from the `music21` corpus. In this collection, each hypergraph corresponds to a music piece, each edge corresponds to a chord sounding for a specific duration at a particular offset from the start of the piece, and each node corresponds to a sound frequency. Note that hypergraphs in the mus collection are node-aligned, which distinguishes them from the hypergraphs in all other collections. In Table 4, we show the cardinality decomposition of selected music hypergraphs that include the largest edges. There, we include edges of cardinality 0 for completeness (they correspond to pauses in the music), but they are discarded in our curvature computations.

**Caveats.** When constructing our hypergraph collection from the `music21` corpus, we excluded pieces that are primarily monophonic. After exploring the corpus manually and evaluating the chord

Table 3: Example record from the data underlying the ndc-ai and ndc-pc hypergraphs.

| Column Name | Record Value |
|---|---|
| product_ndc | 71930-020 |
| active_ingredients_names | [ACETAMINOPHEN, HYDROCODONE BITARTRATE] |
| active_ingredients_strengths | [325 mg/1, 7.5 mg/1] |
| pharm_class | [Opioid Agonist [EPC], Opioid Agonists [MoA]] |
| marketing_category | ANDA |
| dea_schedule | CII |
| finished | True |
| packaging | [{'package_ndc': '71930-020-12', 'description': '100 TABLET in 1 BOTTLE (71930-020-12)', 'marketing_start_date': '20180713', 'sample': False}, {'package_ndc': '71930-020-52', 'description': '500 TABLET in 1 BOTTLE (71930-020-52)', 'marketing_start_date': '20180713', 'sample': False}] |
| dosage_form | TABLET |
| product_type | HUMAN PRESCRIPTION DRUG |
| spl_id | 58b53a57-388e-40d0-9985-048e5af09b0d |
| route | [ORAL] |
| product_id | 71930-020_58b53a57-388e-40d0-9985-048e5af09b0d |
| application_number | ANDA210211 |
| labeler_name | Eywa Pharma Inc |
| generic_name | Hydrocodone Bitartrate and Acetaminophen |
| brand_name | Hydrocodone Bitartrate and Acetaminophen |
| brand_name_base | Hydrocodone Bitartrate and Acetaminophen |
| brand_name_suffix | |
| listing_expiration_date | 2022-12-31 |
| marketing_start_date | 2018-07-13 |
| marketing_end_date | |
| openfda | {'manufacturer_name': ['Eywa Pharma Inc'], 'rxcui': ['856999', '857002', '857005'], 'spl_set_id': ['fcd2b59e-8087-475e-9e6b-911bd846ea96'], 'is_original_packager': [True], 'upc': ['0371930021121', '0371930020124', '0371930019128'], 'unii': ['NO70W886KK', '362O9ITL9D']} |

statistics of individual pieces, we decided to use only music with the following prefixes (corresponding to names of composers or collections): bach, beethoven, chopin, haydn, handel, monteverdi, mozart, palestrina, schumann, schubert, verdi, joplin, trecento, weber. Some pieces are included in several editions (e.g., BWV 190.7, the chorale by Johann Sebastian Bach occupying the first two lines of Table 4, which is included in both the original and an instrumental version).

### A.3.5 STEX: STACKEXCHANGE SITES

StackExchange is a platform hosting Q&A communities also known as sites. Each question is assigned at least one and at most five tags. In the second half of August 2022, we used the StackExchange API to download all questions asked on all StackExchange sites listed on the StackExchange data explorer (https://data.stackexchange.com/), along with their associated tags and other metadata (including question titles and, for smaller sites, also question bodies). From our downloads, we created the stex hypergraph collection, in which each hypergraph corresponds to a StackExchange site, each edge corresponds to a question asked on a site, and each node corresponds to a tag used at least once on a site. Tables 5 to 11 list the basic statistics for each hypergraph from the stex collection.

**Caveats.** While our curvature computations uniformly include only questions asked no later than August 15, midnight GMT, the metadata associated with these questions stems from snapshots at different times in the second half of August 2022. We also excluded stackoverflow.com and math.stackexchange.com from our downloads because they could not be downloaded within one day due to API quota limitations, and ru.stackoverflow.com because it was large but we would not have been able to interpret our results.

Table 4: Selection of hypergraphs from the mus collection. $n$ is the number of nodes, $m$ is the number of edges, and the columns labeled $i$ for $i \in \{0, 1, \ldots, 12\}$ record the number of edges of cardinality $i$ in the hypergraph. Identifiers correspond to abbreviated `music21` identifiers and generally have the shape {composer}-{work identifier}-{suffix}, where *o* stands for *opus*, *m* stands for *movement*, and *inst* stands for *instrumental*.

| | $n$ | $m$ | 0 | 1 | 2 | 3 | 4 | 5 | 6 | 7 | 8 | 9 | 10 | 11 | 12 |
|---|---|---|---|---|---|---|---|---|---|---|---|---|---|---|---|
| bach-bwv190.7-inst | 38 | 233 | 1 | 0 | 0 | 4 | 25 | 60 | 56 | 72 | 9 | 6 | 0 | 0 | 0 |
| bach-bwv190.7 | 38 | 233 | 1 | 0 | 0 | 4 | 25 | 60 | 56 | 72 | 9 | 6 | 0 | 0 | 0 |
| bach-bwv248.23-2 | 35 | 155 | 1 | 0 | 0 | 12 | 45 | 90 | 0 | 3 | 1 | 2 | 1 | 0 | 0 |
| bach-bwv248.42-4 | 38 | 386 | 3 | 1 | 11 | 42 | 147 | 106 | 54 | 14 | 7 | 1 | 0 | 0 | 0 |
| beethoven-o133 | 88 | 5 140 | 236 | 565 | 828 | 1 515 | 1 758 | 168 | 42 | 21 | 5 | 2 | 0 | 0 | 0 |
| beethoven-o18no1-m1 | 70 | 1 979 | 28 | 295 | 165 | 472 | 761 | 244 | 7 | 6 | 0 | 0 | 1 | 0 | 0 |
| beethoven-o18no1-m4 | 77 | 2 669 | 13 | 338 | 438 | 678 | 1 032 | 134 | 33 | 1 | 1 | 1 | 0 | 0 | 0 |
| beethoven-o18no4 | 81 | 4 730 | 95 | 465 | 674 | 977 | 1 940 | 521 | 50 | 3 | 3 | 1 | 1 | 0 | 0 |
| beethoven-o59no1-m4 | 75 | 2 338 | 27 | 80 | 231 | 338 | 1 467 | 168 | 18 | 4 | 4 | 0 | 1 | 0 | 0 |
| beethoven-o59no2-m1 | 86 | 2 338 | 60 | 127 | 398 | 427 | 1 065 | 203 | 18 | 30 | 4 | 5 | 0 | 0 | 1 |
| beethoven-o59no3-m4 | 81 | 3 292 | 19 | 381 | 529 | 734 | 1 219 | 255 | 139 | 14 | 1 | 1 | 0 | 0 | 0 |
| beethoven-o74 | 82 | 6 492 | 112 | 440 | 922 | 1 448 | 2 886 | 538 | 119 | 21 | 5 | 1 | 0 | 0 | 0 |
| monteverdi-madrigal.3.6 | 35 | 480 | 1 | 9 | 40 | 194 | 151 | 76 | 4 | 3 | 1 | 1 | 0 | 0 | 0 |
| schumann-clara-o17-m3 | 63 | 819 | 5 | 12 | 133 | 208 | 151 | 108 | 83 | 74 | 25 | 13 | 5 | 2 | 0 |
| schumann-o41no1-m5 | 72 | 2 410 | 51 | 130 | 208 | 592 | 919 | 366 | 117 | 18 | 2 | 4 | 0 | 2 | 1 |

### A.3.6  SHA: SHAKESPEARE'S PLAYS

The sha collection is a subset of the HYPERBARD dataset recently introduced by Coupette et al. (2022), based on the TEI-encoded XML files of Shakespeare's plays provided by Folger Digital Texts. Here, each hypergraph represents one of Shakespeare's plays, which are categorized into three types: comedy, history, and tragedy. In each hypergraph representing a play, nodes correspond to named characters in the play, and edges correspond to groups of characters simultaneously present on stage. These hypergraphs are documented extensively in the paper introducing the HYPERBARD dataset (Coupette et al., 2022).

### A.3.7  SYN-C, SYN-R, SYN-S: SYNTHETIC HYPERGRAPHS

To generate synthetic hypergraphs, we wrote hypergraph generators extending three well-known graph models to hypergraphs.

(i) For syn-c, we extended the configuration model, which, for undirected graphs, is specified by a degree sequence. Our hypergraph configuration model is specified by a node degree sequence and an edge cardinality sequence.

(ii) For syn-r, we extended the Erdős-Rényi random graph model, which, for undirected graphs, is specified by a number of nodes $n$ and an edge existence probability $p$. Our Erdős-Rényi random hypergraph model is specified by a number of nodes $n$, a number of edges $m$, and the probability $p$ of a one in any cell of the node-to-edge incidence matrix.

(iii) For syn-s, we extended the stochastic block model which, for undirected graphs, is specified by a vector of $c$ community sizes and a $c \times c$ affinity matrix specifying affiliation probabilities between communities. Our hypergraph stochastic block model is specified by a vector of $c_V$ node community sizes, a vector of $c_E$ edge community sizes, and a $c_V \times c_E$ affinity matrix specifying affiliation probabilities between node communities and edge communities.

We used each of our generators to create 250 hypergraphs with identical node count $n$, edge count $m$, and density $c/nm$, where $c$ is the number of filled cells in the node-to-edge incidence matrix.

**Caveats.** Our generators work by pairing node and edge indices, and duplicated (node, edge) index pairs are discarded to generate simple hypergraphs, which can lead to small deviations from the input specification in practice.

Table 5: Basic statistics of hypergraphs derived from StackExchange sites. $n$ is the number of nodes, $m$ is the number of edges, and columns labeled $i \in [5]$ count edges of cardinality $i$.

| | $n$ | $m$ | $n/m$ | 1 | 2 | 3 | 4 | 5 |
|---|---|---|---|---|---|---|---|---|
| 3dprinting | 416 | 4 902 | 0.084863 | 1 003 | 1 617 | 1 367 | 649 | 266 |
| 3dprinting.meta | 45 | 197 | 0.228426 | 65 | 85 | 38 | 5 | 4 |
| academia | 457 | 39 270 | 0.011637 | 6 428 | 11 831 | 11 360 | 6 294 | 3 357 |
| academia.meta | 91 | 1 237 | 0.073565 | 396 | 486 | 249 | 95 | 11 |
| ai | 980 | 10 204 | 0.096041 | 767 | 1 805 | 2 696 | 2 427 | 2 509 |
| ai.meta | 49 | 315 | 0.155556 | 100 | 132 | 67 | 11 | 5 |
| alcohol | 154 | 1 138 | 0.135325 | 415 | 406 | 229 | 56 | 32 |
| alcohol.meta | 28 | 94 | 0.297872 | 28 | 42 | 14 | 8 | 2 |
| android | 1 517 | 56 403 | 0.026896 | 12 890 | 18 313 | 14 406 | 6 996 | 3 798 |
| android.meta | 103 | 996 | 0.103414 | 159 | 447 | 281 | 97 | 12 |
| anime | 1 528 | 12 122 | 0.126052 | 9 510 | 2 215 | 348 | 43 | 6 |
| anime.meta | 83 | 900 | 0.092222 | 234 | 384 | 215 | 56 | 11 |
| apple | 969 | 121 999 | 0.007943 | 15 822 | 34 777 | 37 243 | 22 652 | 11 505 |
| apple.meta | 108 | 1 452 | 0.074380 | 354 | 601 | 393 | 90 | 14 |
| arduino | 445 | 23 616 | 0.018843 | 5 838 | 7 357 | 6 027 | 2 858 | 1 536 |
| arduino.meta | 50 | 255 | 0.196078 | 101 | 110 | 34 | 10 | 0 |
| askubuntu | 3 137 | 393 266 | 0.007977 | 68 310 | 104 529 | 105 601 | 68 907 | 45 919 |
| astronomy | 566 | 12 773 | 0.044312 | 2 781 | 3 812 | 3 284 | 1 777 | 1 119 |
| astronomy.meta | 63 | 339 | 0.185841 | 115 | 93 | 76 | 43 | 12 |
| aviation | 1 024 | 22 701 | 0.045108 | 4 294 | 7 193 | 6 384 | 3 231 | 1 599 |
| aviation.meta | 73 | 752 | 0.097074 | 247 | 295 | 155 | 46 | 9 |
| bicycles | 548 | 18 873 | 0.029036 | 4 884 | 6 267 | 4 652 | 2 097 | 973 |
| bicycles.meta | 74 | 442 | 0.167421 | 150 | 197 | 76 | 15 | 4 |
| bioacoustics | 354 | 287 | 1.233449 | 20 | 50 | 101 | 54 | 62 |
| bioacoustics.meta | 36 | 49 | 0.734694 | 4 | 24 | 16 | 5 | 0 |
| bioinformatics | 490 | 4 998 | 0.098039 | 922 | 1 420 | 1 335 | 782 | 539 |
| bioinformatics.meta | 29 | 112 | 0.258929 | 44 | 53 | 15 | 0 | 0 |
| biology | 745 | 27 348 | 0.027241 | 5 487 | 8 618 | 7 093 | 3 742 | 2 408 |
| biology.meta | 88 | 814 | 0.108108 | 280 | 331 | 145 | 44 | 14 |
| bitcoin | 936 | 28 882 | 0.032408 | 6 677 | 8 927 | 7 432 | 3 766 | 2 080 |
| bitcoin.meta | 58 | 434 | 0.133641 | 142 | 202 | 71 | 16 | 3 |
| blender | 371 | 98 724 | 0.003758 | 31 012 | 30 861 | 22 200 | 9 614 | 5 037 |
| blender.meta | 69 | 716 | 0.096369 | 273 | 291 | 108 | 35 | 9 |
| boardgames | 1 000 | 13 166 | 0.075953 | 9 800 | 2 779 | 500 | 75 | 12 |
| boardgames.meta | 75 | 659 | 0.113809 | 197 | 289 | 144 | 27 | 2 |
| bricks | 202 | 4 220 | 0.047867 | 1 391 | 1 669 | 805 | 266 | 89 |
| bricks.meta | 52 | 211 | 0.246445 | 45 | 95 | 51 | 17 | 3 |
| buddhism | 487 | 7 956 | 0.061212 | 2 381 | 2 357 | 1 730 | 896 | 592 |
| buddhism.meta | 59 | 491 | 0.120163 | 104 | 252 | 94 | 30 | 11 |
| cardano | 285 | 2 248 | 0.126779 | 585 | 664 | 548 | 277 | 174 |
| cardano.meta | 24 | 43 | 0.558140 | 18 | 15 | 10 | 0 | 0 |
| chemistry | 370 | 41 571 | 0.008900 | 9 725 | 14 183 | 10 803 | 4 790 | 2 070 |
| chemistry.meta | 90 | 1 034 | 0.087041 | 250 | 441 | 243 | 88 | 12 |
| chess | 387 | 7 864 | 0.049212 | 1 646 | 2 682 | 2 069 | 985 | 482 |
| chess.meta | 62 | 368 | 0.168478 | 102 | 183 | 72 | 9 | 2 |
| chinese | 166 | 10 298 | 0.016120 | 4 467 | 3 438 | 1 628 | 543 | 222 |
| chinese.meta | 60 | 349 | 0.171920 | 93 | 170 | 67 | 12 | 7 |
| christianity | 1 129 | 14 955 | 0.075493 | 1 739 | 3 571 | 4 205 | 2 967 | 2 473 |
| christianity.meta | 110 | 1 579 | 0.069664 | 593 | 589 | 285 | 88 | 24 |
| civicrm | 507 | 14 324 | 0.035395 | 4 639 | 5 150 | 3 085 | 1 083 | 367 |
| civicrm.meta | 18 | 69 | 0.260870 | 43 | 18 | 6 | 2 | 0 |
| codegolf | 257 | 13 228 | 0.019428 | 1 360 | 4 586 | 4 379 | 2 106 | 797 |
| codegolf.meta | 128 | 2 276 | 0.056239 | 559 | 848 | 549 | 245 | 75 |
| codereview | 1 114 | 76 105 | 0.014638 | 6 306 | 20 542 | 23 777 | 16 106 | 9 374 |
| codereview.meta | 133 | 1 947 | 0.068310 | 190 | 615 | 688 | 345 | 109 |

Table 6: Basic statistics of hypergraphs derived from StackExchange sites (continued). $n$ is the number of nodes, $m$ is the number of edges, and columns labeled $i \in [5]$ count edges of cardinality $i$.

| | $n$ | $m$ | $n/m$ | 1 | 2 | 3 | 4 | 5 |
|---|---|---|---|---|---|---|---|---|
| coffee | 114 | 1 381 | 0.082549 | 492 | 524 | 260 | 78 | 27 |
| coffee.meta | 27 | 90 | 0.300000 | 45 | 30 | 13 | 2 | 0 |
| communitybuilding | 74 | 559 | 0.132379 | 148 | 219 | 112 | 55 | 25 |
| communitybuilding.meta | 27 | 132 | 0.204545 | 36 | 67 | 24 | 4 | 1 |
| computergraphics | 259 | 3 600 | 0.071944 | 883 | 1 024 | 877 | 489 | 327 |
| computergraphics.meta | 34 | 150 | 0.226667 | 55 | 66 | 27 | 2 | 0 |
| conlang | 96 | 448 | 0.214286 | 109 | 204 | 91 | 32 | 12 |
| conlang.meta | 21 | 61 | 0.344262 | 16 | 34 | 7 | 4 | 0 |
| cooking | 834 | 25 877 | 0.032229 | 6 568 | 9 266 | 6 344 | 2 682 | 1 017 |
| cooking.meta | 83 | 866 | 0.095843 | 241 | 410 | 178 | 34 | 3 |
| craftcms | 523 | 13 756 | 0.038020 | 3 738 | 4 912 | 3 410 | 1 263 | 433 |
| craftcms.meta | 20 | 50 | 0.400000 | 22 | 11 | 15 | 1 | 1 |
| crafts | 193 | 2 039 | 0.094654 | 706 | 828 | 397 | 84 | 24 |
| crafts.meta | 49 | 184 | 0.266304 | 40 | 88 | 45 | 11 | 0 |
| crypto | 506 | 27 447 | 0.018436 | 6 448 | 9 056 | 6 960 | 3 283 | 1 700 |
| crypto.meta | 74 | 542 | 0.136531 | 139 | 237 | 127 | 27 | 12 |
| cs | 656 | 44 794 | 0.014645 | 8 624 | 14 332 | 12 644 | 6 336 | 2 858 |
| cs.meta | 86 | 603 | 0.142620 | 90 | 247 | 185 | 68 | 13 |
| cseducators | 210 | 1 080 | 0.194444 | 297 | 378 | 252 | 116 | 37 |
| cseducators.meta | 29 | 146 | 0.198630 | 52 | 68 | 26 | 0 | 0 |
| cstheory | 498 | 11 959 | 0.041642 | 1 653 | 3 384 | 3 495 | 2 052 | 1 375 |
| cstheory.meta | 80 | 608 | 0.131579 | 157 | 262 | 156 | 30 | 3 |
| datascience | 663 | 33 997 | 0.019502 | 4 110 | 8 028 | 9 305 | 6 753 | 5 801 |
| datascience.meta | 51 | 237 | 0.215190 | 80 | 97 | 38 | 16 | 6 |
| dba | 1 197 | 96 887 | 0.012355 | 15 956 | 29 750 | 27 361 | 15 610 | 7 682 |
| dba.meta | 76 | 800 | 0.095000 | 280 | 334 | 140 | 38 | 8 |
| devops | 431 | 5 025 | 0.085771 | 1 070 | 1 647 | 1 340 | 616 | 352 |
| devops.meta | 40 | 144 | 0.277778 | 45 | 63 | 31 | 5 | 0 |
| diy | 919 | 71 007 | 0.012942 | 19 347 | 22 079 | 17 371 | 8 399 | 3 811 |
| diy.meta | 68 | 603 | 0.112769 | 227 | 233 | 118 | 21 | 4 |
| drones | 220 | 731 | 0.300958 | 114 | 240 | 193 | 115 | 69 |
| drones.meta | 28 | 62 | 0.451613 | 11 | 31 | 17 | 3 | 0 |
| drupal | 149 | 86 283 | 0.001727 | 25 218 | 37 599 | 18 867 | 4 075 | 524 |
| drupal.meta | 75 | 1 014 | 0.073964 | 361 | 432 | 186 | 35 | 0 |
| dsp | 509 | 24 850 | 0.020483 | 4 460 | 6 779 | 6 565 | 4 081 | 2 965 |
| dsp.meta | 48 | 307 | 0.156352 | 153 | 108 | 30 | 14 | 2 |
| earthscience | 424 | 6 329 | 0.066993 | 1 111 | 1 778 | 1 698 | 1 094 | 648 |
| earthscience.meta | 54 | 321 | 0.168224 | 100 | 145 | 63 | 12 | 1 |
| ebooks | 180 | 1 466 | 0.122783 | 364 | 489 | 339 | 163 | 111 |
| ebooks.meta | 39 | 99 | 0.393939 | 31 | 37 | 23 | 6 | 2 |
| economics | 494 | 13 690 | 0.036085 | 3 488 | 4 426 | 3 160 | 1 678 | 938 |
| economics.meta | 60 | 444 | 0.135135 | 241 | 151 | 40 | 7 | 5 |
| electronics | 2 318 | 175 731 | 0.013191 | 31 201 | 46 423 | 46 974 | 29 107 | 22 026 |
| electronics.meta | 107 | 1 685 | 0.063501 | 698 | 628 | 282 | 62 | 15 |
| elementaryos | 314 | 8 471 | 0.037068 | 3 043 | 2 910 | 1 669 | 619 | 230 |
| elementaryos.meta | 29 | 107 | 0.271028 | 60 | 28 | 17 | 2 | 0 |
| ell | 533 | 99 970 | 0.005332 | 46 764 | 31 310 | 14 644 | 5 147 | 2 105 |
| ell.meta | 93 | 1 224 | 0.075980 | 448 | 489 | 226 | 52 | 9 |
| emacs | 891 | 23 939 | 0.037220 | 7 561 | 9 371 | 4 980 | 1 590 | 437 |
| emacs.meta | 51 | 216 | 0.236111 | 34 | 112 | 59 | 10 | 1 |
| engineering | 468 | 13 867 | 0.033749 | 3 582 | 4 121 | 3 315 | 1 770 | 1 079 |
| engineering.meta | 47 | 217 | 0.216590 | 71 | 87 | 45 | 10 | 4 |
| english | 984 | 125 848 | 0.007819 | 48 232 | 38 850 | 23 112 | 10 111 | 5 543 |
| english.meta | 182 | 3 589 | 0.050711 | 1 224 | 1 305 | 733 | 249 | 78 |
| eosio | 241 | 2 422 | 0.099505 | 766 | 766 | 533 | 245 | 112 |

Table 7: Basic statistics of hypergraphs derived from StackExchange sites (continued). $n$ is the number of nodes, $m$ is the number of edges, and columns labeled $i \in [5]$ count edges of cardinality $i$.

|  | $n$ | $m$ | $n/m$ | 1 | 2 | 3 | 4 | 5 |
|---|---|---|---|---|---|---|---|---|
| eosio.meta | 19 | 27 | 0.703704 | 6 | 14 | 4 | 2 | 1 |
| es.meta.stackoverflow | 168 | 1 817 | 0.092460 | 310 | 665 | 568 | 230 | 44 |
| es.stackoverflow | 2 960 | 179 452 | 0.016495 | 38 027 | 58 218 | 47 343 | 23 415 | 12 449 |
| esperanto | 99 | 1 592 | 0.062186 | 1 050 | 422 | 96 | 16 | 8 |
| esperanto.meta | 20 | 84 | 0.238095 | 37 | 38 | 9 | 0 | 0 |
| ethereum | 891 | 46 678 | 0.019088 | 8 449 | 12 402 | 12 327 | 7 687 | 5 813 |
| ethereum.meta | 63 | 259 | 0.243243 | 98 | 71 | 59 | 26 | 5 |
| expatriates | 304 | 7 182 | 0.042328 | 1 068 | 2 178 | 2 163 | 1 156 | 617 |
| expatriates.meta | 48 | 157 | 0.305732 | 41 | 72 | 41 | 2 | 1 |
| expressionengine | 603 | 12 447 | 0.048445 | 3 724 | 4 239 | 2 901 | 1 150 | 433 |
| expressionengine.meta | 35 | 123 | 0.284553 | 59 | 49 | 15 | 0 | 0 |
| fitness | 402 | 9 667 | 0.041585 | 2 123 | 2 864 | 2 427 | 1 289 | 964 |
| fitness.meta | 54 | 315 | 0.171429 | 126 | 123 | 57 | 7 | 2 |
| freelancing | 125 | 1 946 | 0.064234 | 632 | 654 | 394 | 177 | 89 |
| freelancing.meta | 33 | 132 | 0.250000 | 36 | 64 | 25 | 5 | 2 |
| french | 324 | 12 413 | 0.026102 | 3 368 | 4 126 | 2 923 | 1 390 | 606 |
| french.meta | 73 | 290 | 0.251724 | 58 | 127 | 80 | 24 | 1 |
| gamedev | 1 096 | 54 182 | 0.020228 | 7 381 | 16 130 | 15 996 | 9 433 | 5 242 |
| gamedev.meta | 78 | 910 | 0.085714 | 300 | 430 | 148 | 27 | 5 |
| gaming | 5 883 | 98 355 | 0.059814 | 72 655 | 20 708 | 4 120 | 758 | 114 |
| gaming.meta | 177 | 4 062 | 0.043575 | 478 | 1 853 | 1 219 | 425 | 87 |
| gardening | 526 | 16 629 | 0.031631 | 3 725 | 5 390 | 4 122 | 2 097 | 1 295 |
| gardening.meta | 60 | 320 | 0.187500 | 95 | 157 | 49 | 17 | 2 |
| genealogy | 465 | 3 572 | 0.130179 | 421 | 742 | 1 037 | 902 | 470 |
| genealogy.meta | 56 | 485 | 0.115464 | 133 | 273 | 70 | 8 | 1 |
| german | 265 | 16 022 | 0.016540 | 6 003 | 5 915 | 2 914 | 927 | 263 |
| german.meta | 69 | 540 | 0.127778 | 177 | 224 | 107 | 30 | 2 |
| gis | 2 829 | 150 205 | 0.018834 | 13 868 | 36 527 | 45 339 | 32 527 | 21 944 |
| gis.meta | 91 | 1 016 | 0.089567 | 174 | 361 | 317 | 125 | 39 |
| graphicdesign | 612 | 34 820 | 0.017576 | 7 542 | 10 789 | 9 364 | 4 821 | 2 304 |
| graphicdesign.meta | 83 | 851 | 0.097532 | 253 | 338 | 187 | 58 | 15 |
| ham | 334 | 4 299 | 0.077692 | 927 | 1 287 | 1 199 | 610 | 276 |
| ham.meta | 45 | 156 | 0.288462 | 39 | 65 | 32 | 18 | 2 |
| hardwarerecs | 246 | 3 945 | 0.062357 | 1 201 | 1 366 | 823 | 378 | 177 |
| hardwarerecs.meta | 42 | 255 | 0.164706 | 81 | 100 | 58 | 16 | 0 |
| hermeneutics | 422 | 12 563 | 0.033591 | 2 819 | 3 720 | 3 074 | 1 772 | 1 178 |
| hermeneutics.meta | 63 | 581 | 0.108434 | 256 | 212 | 84 | 22 | 7 |
| hinduism | 825 | 15 771 | 0.052311 | 2 597 | 4 337 | 3 976 | 2 876 | 1 985 |
| hinduism.meta | 89 | 827 | 0.107618 | 196 | 295 | 200 | 98 | 38 |
| history | 843 | 13 784 | 0.061158 | 2 071 | 3 757 | 3 839 | 2 436 | 1 681 |
| history.meta | 68 | 746 | 0.091153 | 340 | 265 | 107 | 31 | 3 |
| homebrew | 415 | 6 113 | 0.067888 | 1 393 | 1 976 | 1 593 | 803 | 348 |
| homebrew.meta | 50 | 172 | 0.290698 | 67 | 63 | 35 | 4 | 3 |
| hsm | 252 | 3 898 | 0.064649 | 982 | 1 272 | 928 | 464 | 252 |
| hsm.meta | 32 | 146 | 0.219178 | 61 | 44 | 37 | 4 | 0 |
| interpersonal | 280 | 3 890 | 0.071979 | 342 | 1 030 | 1 307 | 790 | 421 |
| interpersonal.meta | 76 | 825 | 0.092121 | 214 | 328 | 205 | 62 | 16 |
| iot | 241 | 2 103 | 0.114598 | 560 | 754 | 504 | 193 | 92 |
| iot.meta | 36 | 136 | 0.264706 | 30 | 74 | 27 | 5 | 0 |
| iota | 148 | 1 023 | 0.144673 | 300 | 352 | 248 | 84 | 39 |
| iota.meta | 18 | 38 | 0.473684 | 10 | 20 | 8 | 0 | 0 |
| islam | 562 | 13 792 | 0.040748 | 3 018 | 4 990 | 3 557 | 1 519 | 708 |
| islam.meta | 103 | 864 | 0.119213 | 240 | 358 | 206 | 47 | 13 |
| italian | 94 | 3 590 | 0.026184 | 1 296 | 1 376 | 636 | 206 | 76 |
| italian.meta | 27 | 151 | 0.178808 | 77 | 57 | 14 | 2 | 1 |

Table 8: Basic statistics of hypergraphs derived from StackExchange sites (continued). $n$ is the number of nodes, $m$ is the number of edges, and columns labeled $i \in [5]$ count edges of cardinality $i$.

| | $n$ | $m$ | $n/m$ | 1 | 2 | 3 | 4 | 5 |
|---|---|---|---|---|---|---|---|---|
| ja.meta.stackoverflow | 74 | 1 115 | 0.066368 | 193 | 386 | 306 | 204 | 26 |
| ja.stackoverflow | 1 145 | 28 785 | 0.039778 | 10 077 | 10 518 | 5 624 | 1 946 | 620 |
| japanese | 354 | 26 365 | 0.013427 | 9 325 | 8 869 | 5 191 | 2 020 | 960 |
| japanese.meta | 75 | 817 | 0.091799 | 270 | 351 | 147 | 43 | 6 |
| joomla | 374 | 7 190 | 0.052017 | 1 289 | 2 221 | 2 058 | 1 072 | 550 |
| joomla.meta | 41 | 150 | 0.273333 | 81 | 46 | 19 | 4 | 0 |
| judaism | 1 264 | 36 511 | 0.034620 | 3 753 | 8 116 | 10 854 | 8 042 | 5 746 |
| judaism.meta | 147 | 1 455 | 0.101031 | 108 | 576 | 489 | 222 | 60 |
| korean | 118 | 1 716 | 0.068765 | 767 | 596 | 264 | 69 | 20 |
| korean.meta | 30 | 80 | 0.375000 | 38 | 28 | 8 | 5 | 1 |
| languagelearning | 216 | 1 287 | 0.167832 | 225 | 466 | 354 | 176 | 66 |
| languagelearning.meta | 52 | 195 | 0.266667 | 31 | 103 | 48 | 12 | 1 |
| latin | 370 | 5 400 | 0.068519 | 1 223 | 1 603 | 1 371 | 797 | 406 |
| latin.meta | 46 | 192 | 0.239583 | 34 | 80 | 49 | 25 | 4 |
| law | 938 | 23 649 | 0.039663 | 4 483 | 7 573 | 6 329 | 3 381 | 1 883 |
| law.meta | 66 | 499 | 0.132265 | 117 | 216 | 120 | 36 | 10 |
| lifehacks | 140 | 2 928 | 0.047814 | 1 024 | 1 052 | 595 | 190 | 67 |
| lifehacks.meta | 59 | 268 | 0.220149 | 65 | 122 | 72 | 6 | 3 |
| linguistics | 605 | 10 003 | 0.060482 | 1 947 | 2 836 | 2 556 | 1 627 | 1 037 |
| linguistics.meta | 59 | 363 | 0.162534 | 118 | 159 | 58 | 23 | 5 |
| literature | 2 335 | 5 614 | 0.415924 | 703 | 1 621 | 2 249 | 830 | 211 |
| literature.meta | 63 | 462 | 0.136364 | 56 | 292 | 99 | 15 | 0 |
| magento | 1 811 | 110 316 | 0.016416 | 15 598 | 28 805 | 32 671 | 20 873 | 12 369 |
| magento.meta | 66 | 575 | 0.114783 | 251 | 227 | 78 | 17 | 2 |
| martialarts | 205 | 2 199 | 0.093224 | 461 | 696 | 529 | 326 | 187 |
| martialarts.meta | 40 | 218 | 0.183486 | 66 | 97 | 46 | 9 | 0 |
| math.meta | 232 | 9 169 | 0.025303 | 1 051 | 3 485 | 2 919 | 1 312 | 402 |
| matheducators | 225 | 3 360 | 0.066964 | 696 | 1 118 | 903 | 435 | 208 |
| matheducators.meta | 57 | 255 | 0.223529 | 64 | 119 | 61 | 8 | 3 |
| mathematica | 705 | 85 069 | 0.008287 | 25 896 | 31 653 | 18 182 | 6 542 | 2 796 |
| mathematica.meta | 75 | 914 | 0.082057 | 416 | 341 | 130 | 25 | 2 |
| mathoverflow.net | 1 530 | 137 735 | 0.011108 | 20 381 | 37 763 | 38 643 | 24 597 | 16 351 |
| mattermodeling | 449 | 2 422 | 0.185384 | 169 | 547 | 668 | 495 | 543 |
| mattermodeling.meta | 61 | 142 | 0.429577 | 25 | 41 | 29 | 37 | 10 |
| mechanics | 1 430 | 25 243 | 0.056649 | 4 196 | 6 245 | 7 592 | 4 673 | 2 537 |
| mechanics.meta | 52 | 387 | 0.134367 | 124 | 182 | 66 | 13 | 2 |
| medicalsciences | 1 435 | 7 586 | 0.189164 | 1 423 | 1 970 | 1 754 | 1 261 | 1 178 |
| medicalsciences.meta | 65 | 501 | 0.129741 | 171 | 191 | 102 | 27 | 10 |
| meta.askubuntu | 196 | 5 698 | 0.034398 | 1 625 | 2 308 | 1 257 | 397 | 111 |
| meta | 1 250 | 97 114 | 0.012871 | 4 599 | 25 289 | 34 007 | 23 233 | 9 986 |
| meta.mathoverflow.net | 133 | 1 687 | 0.078838 | 272 | 601 | 504 | 229 | 81 |
| meta.serverfault | 139 | 2 173 | 0.063967 | 767 | 799 | 463 | 119 | 25 |
| meta.stackoverflow | 622 | 47 387 | 0.013126 | 5 297 | 15 301 | 15 792 | 8 233 | 2 764 |
| meta.superuser | 207 | 5 000 | 0.041400 | 1 010 | 1 914 | 1 474 | 510 | 92 |
| monero | 400 | 4 285 | 0.093349 | 1 193 | 1 424 | 969 | 481 | 218 |
| monero.meta | 23 | 85 | 0.270588 | 40 | 26 | 19 | 0 | 0 |
| money | 1 002 | 36 187 | 0.027690 | 3 788 | 8 036 | 10 340 | 8 450 | 5 573 |
| money.meta | 67 | 672 | 0.099702 | 220 | 260 | 147 | 40 | 5 |
| movies | 4 537 | 21 829 | 0.207843 | 4 857 | 11 430 | 4 546 | 877 | 119 |
| movies.meta | 75 | 1 285 | 0.058366 | 302 | 519 | 391 | 63 | 10 |
| music | 516 | 23 424 | 0.022029 | 4 754 | 7 644 | 6 370 | 3 117 | 1 539 |
| music.meta | 81 | 992 | 0.081653 | 391 | 387 | 166 | 40 | 8 |
| musicfans | 237 | 2 990 | 0.079264 | 1 209 | 1 169 | 465 | 111 | 36 |
| musicfans.meta | 42 | 218 | 0.192661 | 62 | 95 | 38 | 18 | 5 |
| mythology | 303 | 1 953 | 0.155146 | 484 | 723 | 439 | 215 | 92 |

Table 9: Basic statistics of hypergraphs derived from StackExchange sites (continued). $n$ is the number of nodes, $m$ is the number of edges, and columns labeled $i \in [5]$ count edges of cardinality $i$.

| | $n$ | $m$ | $n/m$ | 1 | 2 | 3 | 4 | 5 |
|---|---|---|---|---|---|---|---|---|
| mythology.meta | 35 | 162 | 0.216049 | 43 | 87 | 31 | 1 | 0 |
| networkengineering | 453 | 15 624 | 0.028994 | 2 988 | 4 240 | 3 835 | 2 496 | 2 065 |
| networkengineering.meta | 53 | 375 | 0.141333 | 192 | 115 | 48 | 17 | 3 |
| opendata | 302 | 5 990 | 0.050417 | 1 562 | 2 002 | 1 492 | 670 | 264 |
| opendata.meta | 26 | 180 | 0.144444 | 73 | 76 | 30 | 1 | 0 |
| opensource | 203 | 4 226 | 0.048036 | 845 | 1 442 | 1 094 | 528 | 317 |
| opensource.meta | 53 | 225 | 0.235556 | 35 | 109 | 61 | 19 | 1 |
| or | 255 | 2 865 | 0.089005 | 351 | 809 | 848 | 496 | 361 |
| or.meta | 44 | 114 | 0.385965 | 21 | 61 | 23 | 5 | 4 |
| outdoors | 555 | 5 908 | 0.093940 | 934 | 2 017 | 1 791 | 806 | 360 |
| outdoors.meta | 52 | 512 | 0.101562 | 169 | 276 | 60 | 7 | 0 |
| parenting | 304 | 6 636 | 0.045811 | 1 182 | 2 175 | 1 873 | 1 004 | 402 |
| parenting.meta | 61 | 473 | 0.128964 | 96 | 217 | 125 | 31 | 4 |
| patents | 2 102 | 4 381 | 0.479799 | 1 421 | 1 211 | 879 | 481 | 389 |
| patents.meta | 46 | 167 | 0.275449 | 55 | 69 | 34 | 8 | 1 |
| pets | 289 | 7 874 | 0.036703 | 781 | 2 706 | 2 350 | 1 305 | 732 |
| pets.meta | 62 | 407 | 0.152334 | 60 | 194 | 112 | 26 | 15 |
| philosophy | 606 | 17 915 | 0.033826 | 4 898 | 5 399 | 4 079 | 2 089 | 1 450 |
| philosophy.meta | 61 | 793 | 0.076923 | 355 | 258 | 127 | 38 | 15 |
| photo | 1 156 | 25 961 | 0.044528 | 3 395 | 6 960 | 7 848 | 4 936 | 2 822 |
| photo.meta | 107 | 1 095 | 0.097717 | 289 | 500 | 239 | 60 | 7 |
| physics | 892 | 209 515 | 0.004257 | 21 914 | 42 808 | 53 150 | 45 705 | 45 938 |
| physics.meta | 114 | 3 228 | 0.035316 | 713 | 1 085 | 872 | 403 | 155 |
| pm | 283 | 6 198 | 0.045660 | 1 379 | 1 850 | 1 592 | 870 | 507 |
| pm.meta | 64 | 315 | 0.203175 | 81 | 129 | 73 | 27 | 5 |
| poker | 131 | 2 051 | 0.063871 | 763 | 659 | 372 | 181 | 76 |
| poker.meta | 29 | 122 | 0.237705 | 74 | 30 | 15 | 3 | 0 |
| politics | 793 | 14 628 | 0.054211 | 1 294 | 4 022 | 4 663 | 3 062 | 1 587 |
| politics.meta | 80 | 1 067 | 0.074977 | 249 | 436 | 259 | 103 | 20 |
| portuguese | 169 | 2 349 | 0.071946 | 703 | 898 | 509 | 174 | 65 |
| portuguese.meta | 35 | 137 | 0.255474 | 45 | 61 | 25 | 5 | 1 |
| proofassistants | 223 | 434 | 0.513825 | 80 | 175 | 116 | 42 | 21 |
| proofassistants.meta | 37 | 64 | 0.578125 | 11 | 26 | 18 | 7 | 2 |
| psychology | 401 | 7 641 | 0.052480 | 1 632 | 2 229 | 1 971 | 1 115 | 694 |
| psychology.meta | 62 | 557 | 0.111311 | 199 | 237 | 90 | 25 | 6 |
| pt.meta.stackoverflow | 140 | 2 986 | 0.046885 | 703 | 1 081 | 775 | 362 | 65 |
| pt.stackoverflow | 2 936 | 152 483 | 0.019255 | 28 143 | 50 055 | 42 386 | 21 287 | 10 612 |
| puzzling | 209 | 24 985 | 0.008365 | 6 912 | 9 471 | 5 731 | 2 020 | 851 |
| puzzling.meta | 98 | 1 365 | 0.071795 | 351 | 582 | 309 | 97 | 26 |
| quant | 693 | 20 283 | 0.034167 | 3 329 | 5 345 | 5 392 | 3 556 | 2 661 |
| quant.meta | 47 | 252 | 0.186508 | 95 | 115 | 37 | 3 | 2 |
| quantumcomputing | 306 | 7 823 | 0.039115 | 1 124 | 2 585 | 2 475 | 1 105 | 534 |
| quantumcomputing.meta | 50 | 187 | 0.267380 | 50 | 73 | 43 | 18 | 3 |
| raspberrypi | 598 | 35 872 | 0.016670 | 7 901 | 11 252 | 9 351 | 4 765 | 2 603 |
| raspberrypi.meta | 61 | 451 | 0.135255 | 213 | 169 | 58 | 8 | 3 |
| retrocomputing | 546 | 4 976 | 0.109727 | 925 | 1 694 | 1 366 | 692 | 299 |
| retrocomputing.meta | 70 | 304 | 0.230263 | 30 | 188 | 56 | 27 | 3 |
| reverseengineering | 347 | 8 754 | 0.039639 | 1 878 | 2 693 | 2 172 | 1 249 | 762 |
| reverseengineering.meta | 37 | 150 | 0.246667 | 56 | 62 | 28 | 3 | 1 |
| robotics | 276 | 6 261 | 0.044082 | 1 528 | 1 850 | 1 519 | 806 | 558 |
| robotics.meta | 39 | 159 | 0.245283 | 52 | 71 | 28 | 8 | 0 |
| rpg | 1 247 | 46 635 | 0.026740 | 4 236 | 12 463 | 15 431 | 9 542 | 4 963 |
| rpg.meta | 150 | 2 627 | 0.057099 | 310 | 986 | 844 | 379 | 108 |
| ru.meta.stackoverflow | 242 | 4 613 | 0.052460 | 445 | 1 312 | 1 574 | 979 | 303 |
| rus | 390 | 20 999 | 0.018572 | 12 276 | 5 131 | 2 341 | 840 | 411 |

Table 10: Basic statistics of hypergraphs derived from StackExchange sites (continued). $n$ is the number of nodes, $m$ is the number of edges, and columns labeled $i \in [5]$ count edges of cardinality $i$.

| | $n$ | $m$ | $n/m$ | 1 | 2 | 3 | 4 | 5 |
|---|---|---|---|---|---|---|---|---|
| rus.meta | 30 | 214 | 0.140187 | 92 | 81 | 37 | 4 | 0 |
| russian | 166 | 4 516 | 0.036758 | 2 407 | 1 337 | 552 | 180 | 40 |
| russian.meta | 37 | 176 | 0.210227 | 80 | 61 | 25 | 7 | 3 |
| salesforce | 2 085 | 124 492 | 0.016748 | 22 537 | 37 977 | 33 635 | 19 220 | 11 123 |
| salesforce.meta | 79 | 795 | 0.099371 | 412 | 246 | 118 | 18 | 1 |
| scicomp | 346 | 10 381 | 0.033330 | 1 905 | 3 156 | 2 883 | 1 566 | 871 |
| scicomp.meta | 48 | 215 | 0.223256 | 75 | 90 | 42 | 8 | 0 |
| scifi | 3 693 | 69 344 | 0.053256 | 17 338 | 26 498 | 17 146 | 6 584 | 1 778 |
| scifi.meta | 149 | 3 265 | 0.045636 | 506 | 1 560 | 889 | 266 | 44 |
| security | 1 253 | 65 817 | 0.019038 | 11 950 | 19 799 | 18 266 | 9 809 | 5 993 |
| security.meta | 101 | 1 124 | 0.089858 | 311 | 507 | 242 | 52 | 12 |
| serverfault | 3 864 | 314 342 | 0.012292 | 40 967 | 83 417 | 92 763 | 60 560 | 36 635 |
| sharepoint | 1 722 | 99 911 | 0.017235 | 16 092 | 27 312 | 28 073 | 17 305 | 11 129 |
| sharepoint.meta | 78 | 581 | 0.134251 | 206 | 233 | 127 | 14 | 1 |
| sitecore | 362 | 11 395 | 0.031768 | 5 106 | 4 265 | 1 611 | 342 | 71 |
| sitecore.meta | 24 | 202 | 0.118812 | 40 | 60 | 99 | 3 | 0 |
| skeptics | 682 | 10 700 | 0.063738 | 2 227 | 4 165 | 2 952 | 1 042 | 314 |
| skeptics.meta | 100 | 1 529 | 0.065402 | 528 | 605 | 310 | 77 | 9 |
| softwareengineering | 1 674 | 61 392 | 0.027267 | 8 950 | 17 773 | 17 580 | 10 572 | 6 517 |
| softwareengineering.meta | 165 | 2 611 | 0.063194 | 421 | 1 023 | 776 | 310 | 81 |
| softwarerecs | 962 | 21 792 | 0.044145 | 3 090 | 6 533 | 6 199 | 3 723 | 2 247 |
| softwarerecs.meta | 85 | 654 | 0.129969 | 86 | 297 | 189 | 66 | 16 |
| sound | 1 224 | 9 786 | 0.125077 | 2 122 | 2 717 | 2 330 | 1 624 | 993 |
| sound.meta | 42 | 160 | 0.262500 | 65 | 66 | 25 | 1 | 3 |
| space | 1 203 | 17 392 | 0.069170 | 1 672 | 4 012 | 4 924 | 3 712 | 3 072 |
| space.meta | 74 | 682 | 0.108504 | 205 | 237 | 150 | 63 | 27 |
| spanish | 274 | 8 592 | 0.031890 | 2 276 | 2 722 | 2 140 | 1 010 | 444 |
| spanish.meta | 84 | 498 | 0.168675 | 94 | 216 | 135 | 42 | 11 |
| sports | 261 | 5 730 | 0.045550 | 926 | 2 371 | 1 637 | 609 | 187 |
| sports.meta | 57 | 350 | 0.162857 | 76 | 170 | 82 | 21 | 1 |
| sqa | 462 | 11 242 | 0.041096 | 2 263 | 3 250 | 2 881 | 1 705 | 1 143 |
| sqa.meta | 41 | 211 | 0.194313 | 115 | 71 | 17 | 7 | 1 |
| stackapps | 210 | 2 756 | 0.076197 | 277 | 858 | 883 | 514 | 224 |
| stats | 1 572 | 196 835 | 0.007986 | 19 622 | 47 967 | 57 502 | 41 443 | 30 301 |
| stats.meta | 132 | 1 685 | 0.078338 | 327 | 576 | 491 | 198 | 93 |
| stellar | 115 | 1 493 | 0.077026 | 585 | 438 | 298 | 109 | 63 |
| stellar.meta | 19 | 31 | 0.612903 | 9 | 14 | 8 | 0 | 0 |
| substrate | 512 | 1 814 | 0.282249 | 366 | 563 | 491 | 260 | 134 |
| substrate.meta | 40 | 44 | 0.909091 | 6 | 21 | 13 | 2 | 2 |
| superuser | 5 676 | 480 854 | 0.011804 | 64 273 | 127 561 | 135 549 | 91 137 | 62 334 |
| sustainability | 234 | 2 012 | 0.116302 | 431 | 713 | 536 | 235 | 97 |
| sustainability.meta | 37 | 151 | 0.245033 | 38 | 75 | 32 | 6 | 0 |
| tex | 2 035 | 237 763 | 0.008559 | 60 247 | 84 998 | 59 476 | 23 747 | 9 295 |
| tex.meta | 163 | 2 277 | 0.071585 | 389 | 921 | 671 | 235 | 61 |
| tezos | 210 | 1 828 | 0.114880 | 567 | 605 | 380 | 180 | 96 |
| tezos.meta | 18 | 32 | 0.562500 | 7 | 15 | 8 | 1 | 1 |
| tor | 218 | 5 636 | 0.038680 | 1 888 | 1 817 | 1 147 | 464 | 320 |
| tor.meta | 43 | 163 | 0.263804 | 57 | 76 | 25 | 4 | 1 |
| travel | 1 916 | 45 040 | 0.042540 | 2 985 | 8 914 | 13 809 | 11 528 | 7 804 |
| travel.meta | 99 | 1 379 | 0.071791 | 293 | 567 | 406 | 98 | 15 |
| tridion | 274 | 7 234 | 0.037877 | 1 471 | 2 758 | 1 915 | 818 | 272 |
| tridion.meta | 14 | 138 | 0.101449 | 93 | 39 | 6 | 0 | 0 |
| ukrainian | 124 | 2 094 | 0.059217 | 664 | 873 | 404 | 127 | 26 |
| ukrainian.meta | 33 | 104 | 0.317308 | 21 | 45 | 31 | 6 | 1 |
| unix | 2 777 | 220 644 | 0.012586 | 29 059 | 61 964 | 66 657 | 40 340 | 22 624 |

Table 11: Basic statistics of hypergraphs derived from StackExchange sites (continued). $n$ is the number of nodes, $m$ is the number of edges, and columns labeled $i \in [5]$ count edges of cardinality $i$.

|  | $n$ | $m$ | $n/m$ | 1 | 2 | 3 | 4 | 5 |
|---|---|---|---|---|---|---|---|---|
| unix.meta | 118 | 1 668 | 0.070743 | 367 | 727 | 407 | 144 | 23 |
| ux | 1 032 | 31 459 | 0.032805 | 4 660 | 8 934 | 8 823 | 5 530 | 3 512 |
| ux.meta | 94 | 899 | 0.104561 | 273 | 358 | 199 | 54 | 15 |
| vegetarianism | 115 | 677 | 0.169867 | 85 | 233 | 205 | 106 | 48 |
| vegetarianism.meta | 41 | 133 | 0.308271 | 26 | 62 | 32 | 13 | 0 |
| vi | 421 | 12 558 | 0.033524 | 4 494 | 4 802 | 2 358 | 694 | 210 |
| vi.meta | 35 | 201 | 0.174129 | 63 | 105 | 30 | 3 | 0 |
| video | 327 | 8 661 | 0.037755 | 2 705 | 2 693 | 1 831 | 882 | 550 |
| video.meta | 41 | 200 | 0.205000 | 63 | 96 | 32 | 8 | 1 |
| webapps | 951 | 33 202 | 0.028643 | 14 343 | 11 667 | 5 160 | 1 435 | 597 |
| webapps.meta | 106 | 937 | 0.113127 | 97 | 447 | 311 | 76 | 6 |
| webmasters | 1 078 | 36 840 | 0.029262 | 5 772 | 10 197 | 10 531 | 6 286 | 4 054 |
| webmasters.meta | 70 | 649 | 0.107858 | 202 | 258 | 135 | 45 | 9 |
| windowsphone | 287 | 3 440 | 0.083430 | 975 | 1 257 | 801 | 306 | 101 |
| windowsphone.meta | 44 | 148 | 0.297297 | 47 | 64 | 27 | 8 | 2 |
| woodworking | 244 | 3 739 | 0.065258 | 1 129 | 1 270 | 880 | 347 | 113 |
| woodworking.meta | 34 | 142 | 0.239437 | 69 | 46 | 25 | 2 | 0 |
| wordpress | 702 | 112 778 | 0.006225 | 27 669 | 37 039 | 28 491 | 13 228 | 6 351 |
| wordpress.meta | 82 | 866 | 0.094688 | 381 | 330 | 118 | 30 | 7 |
| workplace | 498 | 30 369 | 0.016398 | 6 371 | 9 325 | 8 103 | 4 221 | 2 349 |
| workplace.meta | 113 | 1 829 | 0.061782 | 506 | 699 | 447 | 150 | 27 |
| worldbuilding | 675 | 34 358 | 0.019646 | 2 958 | 8 284 | 10 839 | 7 267 | 5 010 |
| worldbuilding.meta | 120 | 2 032 | 0.059055 | 445 | 901 | 511 | 147 | 28 |
| writing | 391 | 11 699 | 0.033422 | 2 456 | 3 869 | 3 055 | 1 557 | 762 |
| writing.meta | 88 | 789 | 0.111534 | 145 | 415 | 173 | 49 | 7 |

## A.4 IMPLEMENTATION DETAILS

To simplify the computation of Wasserstein distances between adjacent nodes, we leverage the following fact about the relevant distances (i.e., transportation costs) between nodes.

**Lemma 1.** *Given a hypergraph $H = (V, E)$ and nodes $i, j, k, \ell \in V$ with $i \sim j$ as well as $\mu_i(k) > 0$ and $\mu_j(\ell) > 0$, $\mathrm{d}(k, \ell) \leq 3$.*

*Proof.* By the triangle inequality and the definition of our probability measures, we have

$$\mathrm{d}(k, \ell) \leq \mathrm{d}(k, i) + \mathrm{d}(i, j) + \mathrm{d}(j, \ell) = 3 .$$

$\square$

Furthermore, we speed up the computation of Wasserstein distances by exploiting the following observation to reduce each instance to its smallest equivalent instance.

**Lemma 2.** *Given a hypergraph $H = (V, E)$ and nodes $i, j \in V$ with $i \sim j$, if $\mu_i(k) = \mu_j(k)$ for some node $k \in V$, then $\mathrm{W}_1(\mu_i, \mu_j) = \mathrm{W}_1(\mu_i^{-k}, \mu_j^{-k})$, where $\mu_i^{-k}$ is defined as*

$$\mu_i^{-k}(j) := \begin{cases} 0 & j = k \\ \mu_i(j) & j \neq k . \end{cases}$$

*Proof.* If $\mu_i(k) = \mu_j(k) = 0$, the claim holds trivially. Otherwise, $\mu_i(k) = \mu_j(k) = \beta > 0$. In this case, let $C^*$ be an optimal coupling between $\mu_i$ and $\mu_j$. If the probability mass allocated to $k$ by $\mu_i$ does not get moved at all in $C^*$, it contributes 0 to $\mathrm{W}_1(\mu_i, \mu_j)$, and we are done. Therefore, assume otherwise. Then there exist nodes $p, q \in V$ such that probability mass gets moved from $p$ to $k$ and from $k$ to $q$ in $C^*$. By the triangle inequality, $\mathrm{d}(p, q) \leq \mathrm{d}(p, k) + \mathrm{d}(k, q)$, and as $\mathrm{d}(k, k) = 0$, the cost of moving that mass directly from $p$ to $q$ and keeping all mass at $k$ cannot be larger than the cost of moving the mass from $p$ to $k$ and from $k$ to $q$. Hence, we can modify $C^*$ such that the mass allocated to $k$ by $\mu_i$ does not get moved at all without increasing the coupling cost. Thus, there always exists an optimal coupling in which all mass at $k$ remains at $k$, and the claim follows. $\square$

### A.5  FURTHER RESULTS

Here, we showcase further results to support and supplement the exposition in the main paper.

**Q1 Parametrization.**  Expanding the discussion on ORCHID parametrizations, Fig. 5 shows the distributions of edge curvatures and edge-averaged node curvatures for two hypergraphs from the dblp-v collection, representing top conferences in machine learning and theoretical computer science, respectively. The figure highlights once more the consistently concentrating effect of increasing $\alpha$, and it elucidates the differential effects of moving from maximum aggregation (left parts of the split violins) to mean aggregation (right parts of the split violins), from almost no shifts to large shifts in probability mass (compare, e.g., Fig. 5b, top right panel, with Fig. 5b, bottom left panel). Fig. 5 might convey the impression that, other parameters being equal, the distributions of curvatures based on $\mu^{EN}$ and $\mu^{WE}$ are more similar to each other than to $\mu^{EE}$. This does not hold in general, however, as demonstrated for ndc-pc in Fig. 6a, where node curvature distributions based on $\mu^{WE}$ are more similar to those based on $\mu^{E\bar{E}}$ than to the node curvature distributions based on $\mu^{EN}$. Comparing Fig. 6a to Fig. 6b (ndc-ai), we further observe that rather similar distributions of edge curvature and directional curvature can be accompanied by rather different distributions of edge-averaged and direction-averaged node curvatures, even for hypergraphs originating from the same domain. Finally, when visualizing curvatures for hypergraphs in the same collection or across collections with related semantics (Fig. 7), we can identify several distinct prototypical shapes of curvature distributions and relationships between curvatures based on different probability measures.

**Q2 Hypergraph Exploration.**  Extending the discussion of individual hypergraph exploration in the main paper, we focus on a case study of the citation hypergraph of the journal Physical Review E (PRE), which regularly publishes, inter alia, interdisciplinary work on graphs and networks. In this hypergraph, which has 45 504 nodes and 52 574 edges, nodes represent PRE articles *cited* by at least one other PRE article, edges represent PRE articles *citing* at least one other PRE article, and each edge $i$ comprises the nodes $j$ cited by the paper corresponding to $i$. Therefore, the *edge* curvature of a (citing) paper $i$ can be interpreted as an indicator of its *breadth of content*: The more *positive* the edge curvature, the stronger the general tendency of the papers jointly cited by paper $i$ to be cited together, suggesting that these papers are topically related. Similarly, the *node* curvature of a (cited) paper $j$ can be interpreted as an indicator of its *breadth of impact*: The more *negative* the node curvature, the more diversely the paper has been cited in the literature.

With these interpretations in mind, we compute all curvatures for the PRE citation hypergraph, using $\alpha = 0.1$, $\mu^{WE}$, and $\mathtt{AGG_A}$. We find that for all 54 articles with at least 100 citations (top articles), the edge-averaged node curvature is larger than the direction-averaged node curvature, which is always negative, although only 36% of all PRE articles exhibit this feature combination. This matches the intuition that from highly cited articles, the literature should diverge in many different directions. At the same time, we observe that curvatures span a considerable range, even among top articles. In Table 12, we record the top articles with extreme curvature values, and in Fig. 8, we display the pairwise relationships between curvature features and other local features for *all* PRE articles. In line with the interpretations sketched above, the top article with the largest node curvatures is a classic reference for community detection in the highly integrated field of network science, whereas the articles with the smallest node curvatures address topics relevant to a broader range of approaches to collective phenomena in many-body systems (which are the focus of PRE).

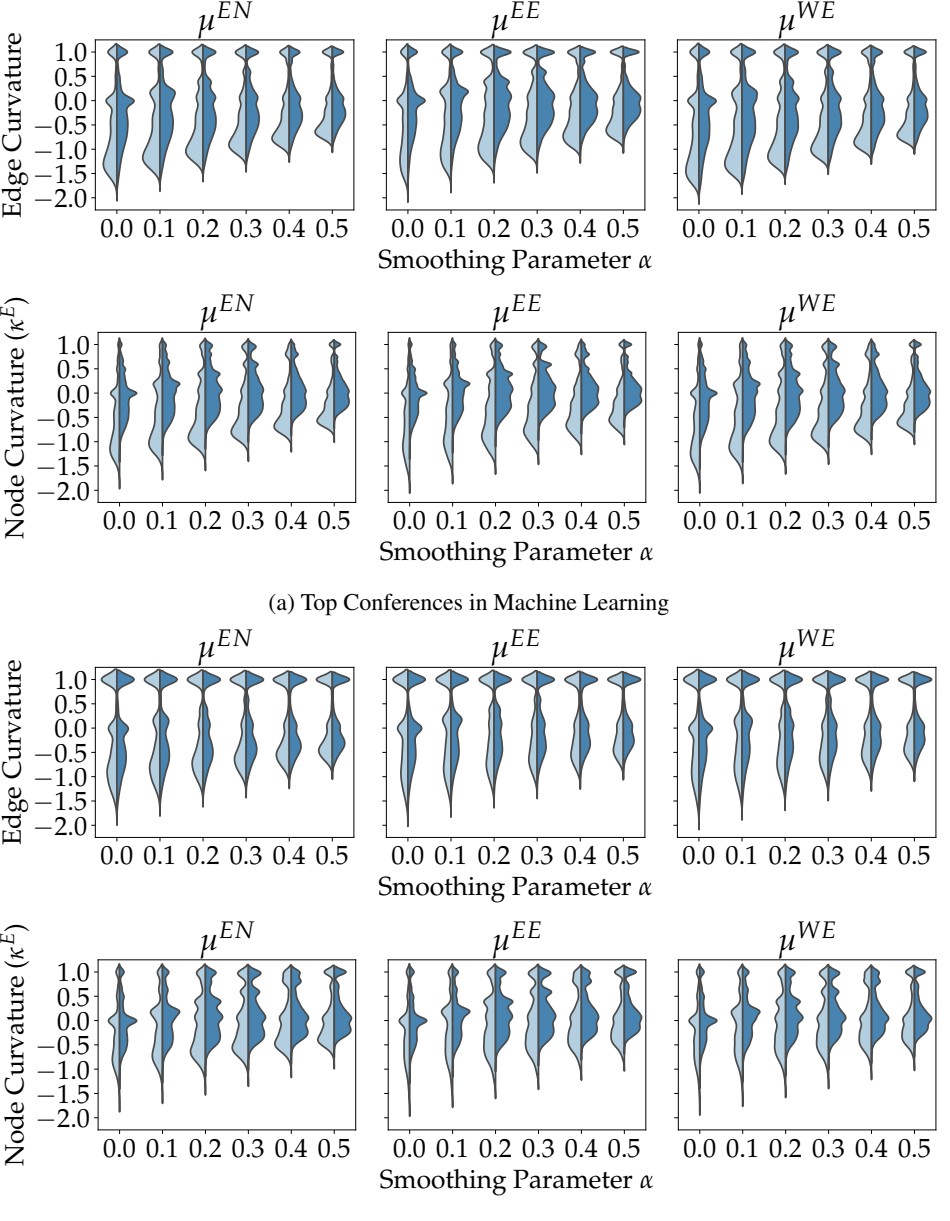

(a) Top Conferences in Machine Learning

(b) Top Conferences in Theoretical Computer Science

Figure 5: ORCHID curvatures are non-redundant. We show distributions of ORCHID edge curvatures (top) and edge-averaged node curvatures (bottom) using probability measures $\mu^{EN}$, $\mu^{EE}$, and $\mu^{WE}$ with smoothing $\alpha$, for the aggregation functions $\text{AGG}_M$ (light blue) and $\text{AGG}_A$ (dark blue) on dblp-v hypergraphs representing top conferences in machine learning and in theoretical computer science.

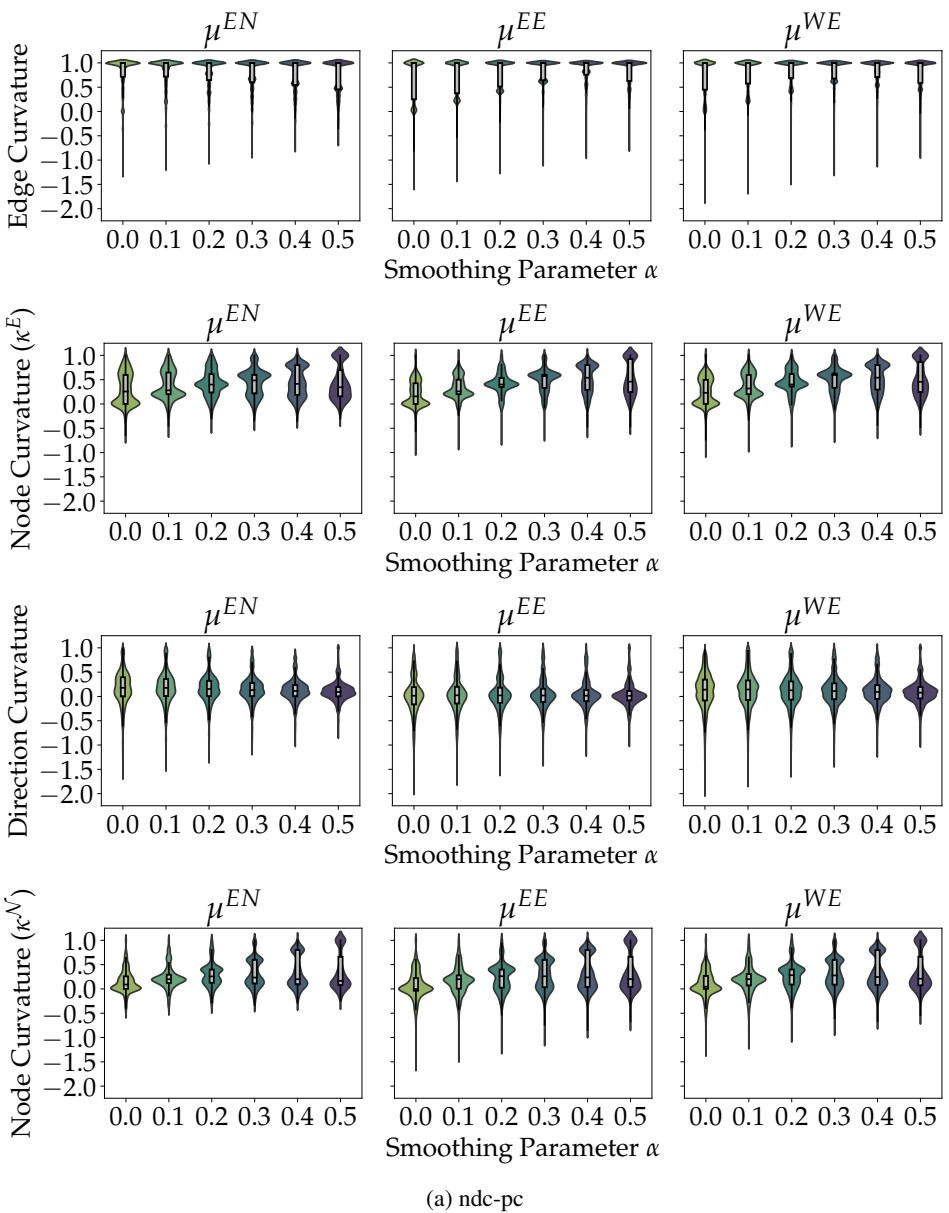

(a) ndc-pc

Figure 6: Hypergraphs with similar distributions of one curvature type may differ in their distributions of other curvature types. We show ORCHID curvatures computed using AGG$_A$, for all curvature types, probability measures, and $\alpha \in \{0.0, 0.1, 0.2, 0.3, 0.4, 0.5\}$. (Figure continues on next page.)

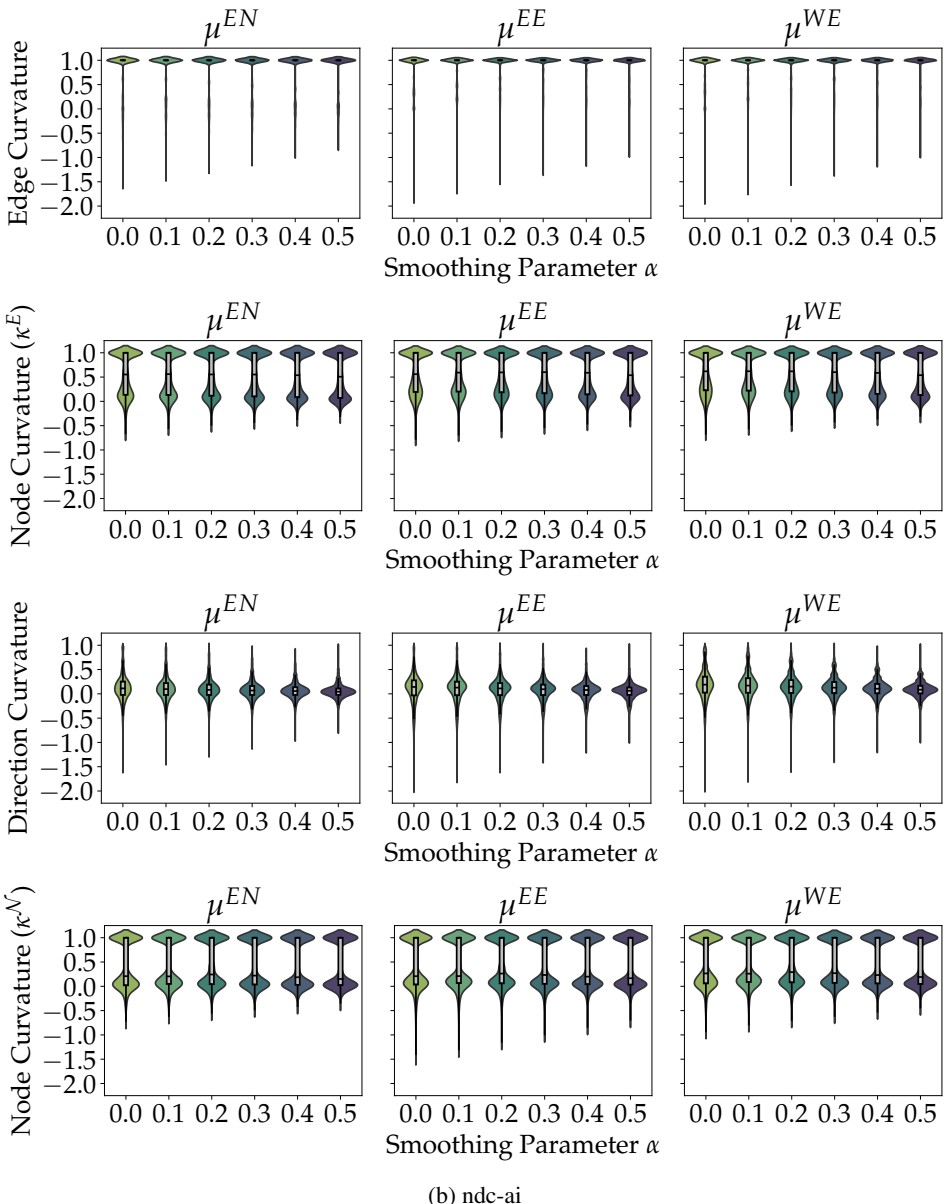

(b) ndc-ai

Figure 6: Hypergraphs with similar distributions of one curvature type may differ in their distributions of other curvature types. We show ORCHID curvatures computed using AGG$_A$, for all curvature types, probability measures, and $\alpha \in \{0.0, 0.1, 0.2, 0.3, 0.4, 0.5\}$. (Figure continued from previous page.)

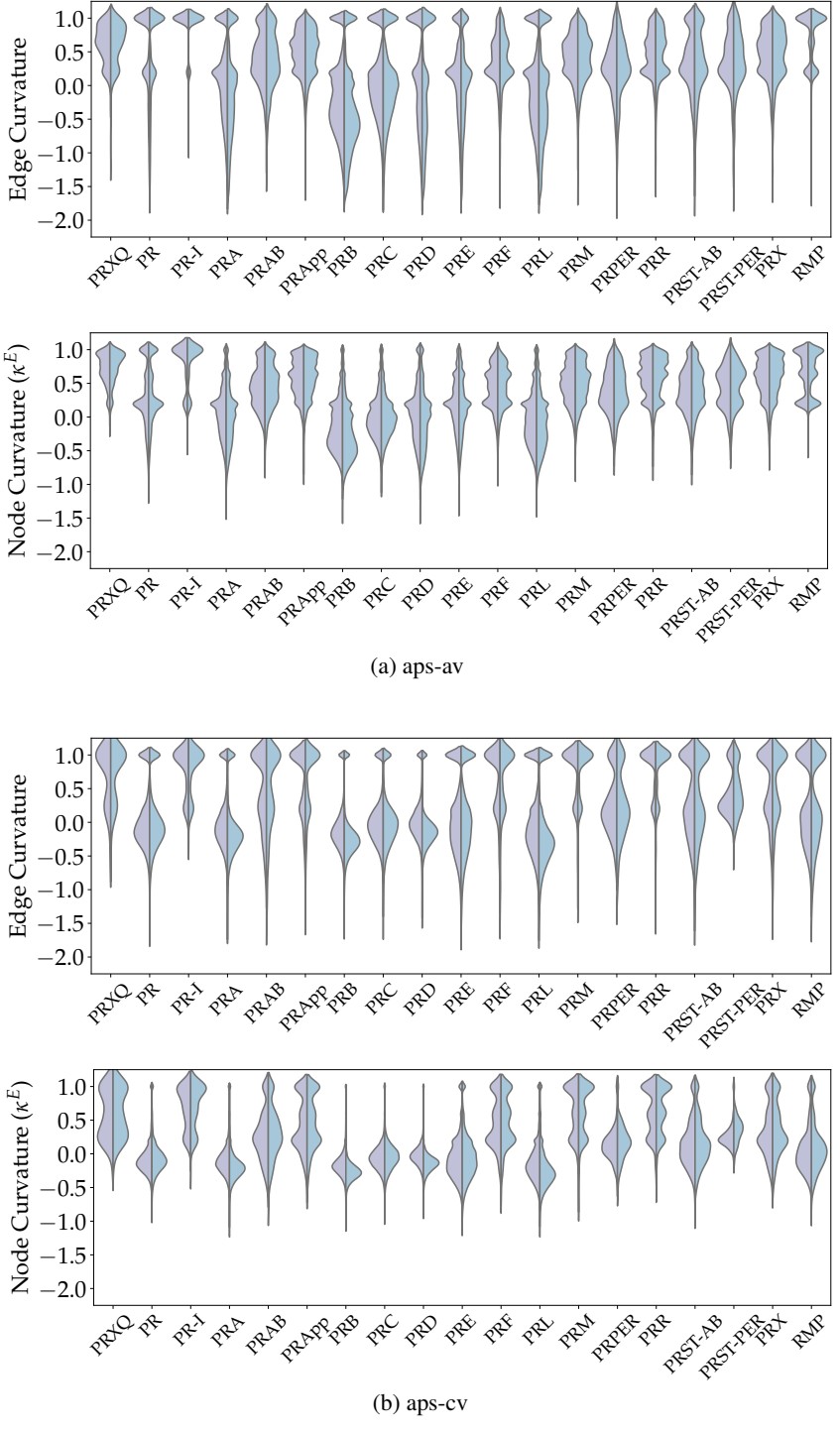

(a) aps-av

(b) aps-cv

Figure 7: ORCHID curvature distributions within the same collection and across semantically related collections exhibit prototypical shapes, accompanied by varying types of relationships between probability measures. We show distributions of ORCHID edge curvatures (top) and edge-averaged node curvatures (bottom) computed using $\alpha = 0.1$ and $\mathrm{A}\mathtt{GG}_{\mathtt{A}}$, for $\mu^{\mathrm{EE}}$ (violet) and $\mu^{\mathrm{WE}}$ (blue), for all hypergraphs in aps-av and aps-cv. Recall that the edges in aps-av and aps-cv as well as the nodes in aps-cv represent essentially the same set of APS papers, but in aps-av, they connect co-authors, and in aps-cv, they connect co-cited papers (edges) or are connected by citing papers (nodes).

Table 12: Top articles display varying relationships between different curvature values. We list the PRE articles that, out of all PRE articles cited at least 100 times, exhibit the most extreme curvature-related values.

| | DOI | $\kappa^E(i)$ | $\kappa^{\mathcal{N}}(i)$ | $\Delta(\kappa(i))$ | $\kappa(e)$ | Title |
|---|---|---|---|---|---|---|
| max $\kappa^E(i)$, max $\kappa^{\mathcal{N}}(i)$ | 10.1103/PhysRevE.70.066111 | 0.220092 | -0.006001 | 0.226093 | 0.425336 | Finding community structure in very large networks |
| min $\kappa^E(i)$ | 10.1103/PhysRevE.47.851 | -0.319638 | -0.555431 | 0.235793 | 0 | Scale-invariant motion in intermittent chaotic systems |
| min $\kappa^{\mathcal{N}}(i)$ | 10.1103/PhysRevE.48.R29 | -0.241216 | -0.704752 | 0.463536 | 0 | Extended self-similarity in turbulent flows |
| max $\Delta(\kappa(i))$ | 10.1103/PhysRevE.64.056101 | -0.131542 | -0.668266 | 0.536724 | 0.038477 | Determining the density of states for classical statistical models: A random walk algorithm to produce a flat histogram |
| min $\Delta(\kappa(i))$ | 10.1103/PhysRevE.74.016118 | -0.015495 | -0.191193 | 0.175697 | -0.156824 | Amorphous systems in athermal, quasistatic shear |
| max $\kappa(e)$ | 10.1103/PhysRevE.57.610 | 0.129557 | -0.251635 | 0.381192 | 0.610123 | Topological defects and interactions in nematic emulsions |
| min $\kappa(e)$ | 10.1103/PhysRevE.64.016706 | -0.191094 | -0.552908 | 0.361815 | -0.644446 | Fast Monte Carlo algorithm for site or bond percolation |

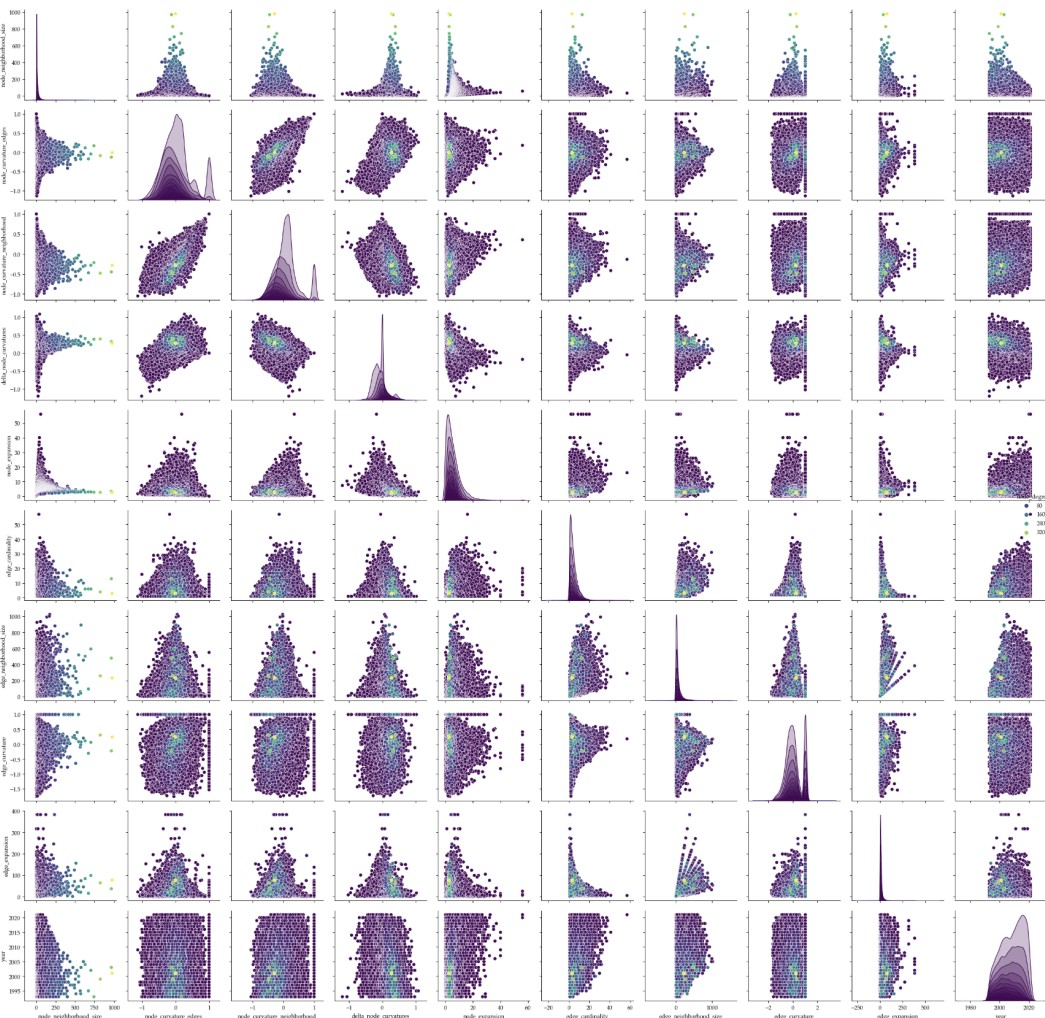

Figure 8: Highly cited articles have distinct curvature distributions. Pairwise relationships between (left-to-right, top-to-bottom) node neighborhood size, edge-averaged node curvature, direction-averaged node curvature, curvature delta, node expansion $:= {}^{\deg(i)}/|\mathcal{N}(i)|$, edge cardinality, edge neighborhood size, edge curvature, edge expansion $:= {}^{\deg(e)}/|\mathcal{N}(e)|$, and (as an additional metadata feature) publication year, for all PRE articles cited at least once by another PRE article, colored by node degree (number of citations within PRE), where brighter colors signal larger node degrees.

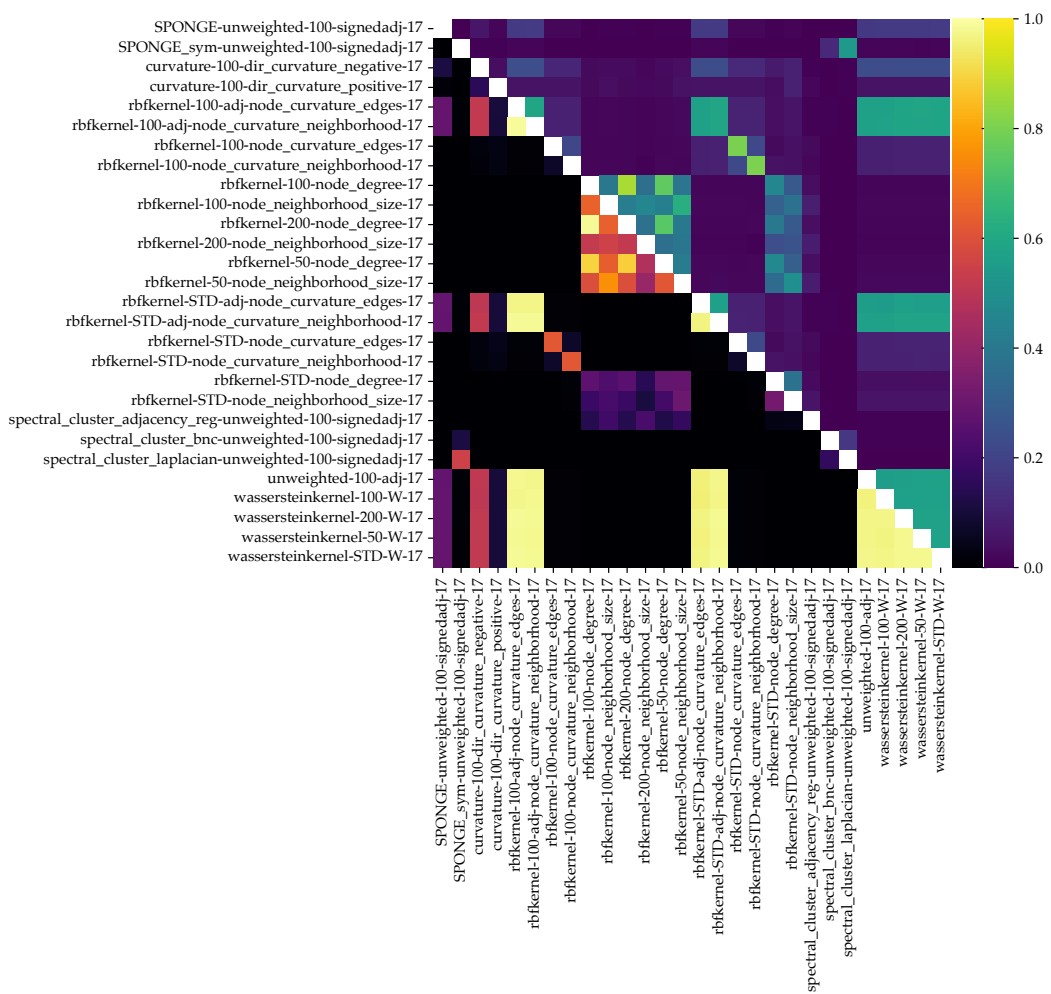

Figure 9: Node clusterings based on curvature features differ radically from clusterings based on other local features. We show the normalized mutual information (upper triangle) and the adjusted rand score (lower triangle) of node clusterings based on different method/feature combinations, computed on the citation hypergraph of PRB from the aps-cv collection, with curvatures computed using $\alpha = 0.1$, $\mu^{\text{WE}}$, and $\text{AGG}_{\text{A}}$.

**Q3 Hypergraph Learning.**    Continuing the discussion of node clustering in hypergraphs abridged in the main paper, we again focus on the citation hypergraph corresponding to articles from Physical Review E (PRE). We experiment with a variety of features, clustering methods, and combinations thereof, including both classic and recent clustering methods, such as SPONGE (Cucuringu et al., 2019). We aim for 17 clusters, which is the number of "disciplines" present in the APS metadata (unfortunately, disciplines are only assigned to more recent articles, and hence, cannot serve as ground truth). As depicted in Fig. 9, we find that clusterings generated using curvatures as features differ radically from clusterings generated using other local features. To evaluate the semantic sensibility of our clusterings in the absence of a suitable ground truth, we leverage the metadata associated with PRE articles. In particular, we concatenate the titles of the articles grouped in each of our clusters into "documents", and consider the set of all clusters as our "document collection", to then identify characteristic terms for each cluster using TF-IDF feature extraction. We observe that clusterings based on ORCHID features tend to be more thematically coherent than clusterings based on other local features. As illustrated in Table 13, ORCHID features tend to separate paper titles well by topic (many frequently occurring terms are associated with only very few clusters, and the terms grouped together characterize specific subfields of the physics of collective phenomena covered by PRE), whereas clusters based on non-ORCHID features are much less topically focused.

Table 13: ORCHID features lead to node clusterings that are semantically more coherent than node clusterings derived from other local features. For two clusterings of the PRE citation hypergraph from the aps-cv collection—one a spectral clustering using the sign of directional curvatures as a feature (Table 13a), the other a clustering using an RBF kernel with node neighborhood size as a feature (Table 13b)—we show the top terms, i.e., the terms associated with each cluster that have a TF-IDF score of at least 0.1, along with their TF-IDF scores and their occurrence frequency across all clusters, in tuples of shape (term, TF-IDF score, global occurrence frequency).

(a) Feature: sign of directional ORCHID curvatures

(smectic, 0.51, 1), (liquid, 0.39, 4), (crystals, 0.22, 4), (antiferroelectric, 0.21, 1), (crystal, 0.19, 2), (phase, 0.17, 4), (chiral, 0.17, 1), (cα, 0.15, 1), (paper, 0.15, 1), (rock, 0.15, 1), (scissors, 0.15, 1), (electric, 0.14, 1), (phases, 0.14, 2), (ray, 0.13, 1), (cyclic, 0.13, 1), (species, 0.12, 1), (field, 0.11, 3), (games, 0.1, 2)

(resetting, 0.76, 1), (stochastic, 0.32, 1), (random, 0.24, 1), (walks, 0.18, 1), (diffusion, 0.17, 2), (brownian, 0.15, 1), (processes, 0.11, 1)

(nematic, 0.66, 2), (liquid, 0.41, 4), (crystal, 0.3, 2), (colloidal, 0.26, 1), (colloids, 0.18, 1), (crystals, 0.16, 4), (particles, 0.15, 1), (interaction, 0.14, 1)

(boltzmann, 0.75, 1), (lattice, 0.51, 1), (method, 0.2, 1), (flows, 0.15, 1), (model, 0.11, 5)

(quantum, 0.58, 3), (heat, 0.38, 1), (engine, 0.34, 1), (engines, 0.27, 1), (efficiency, 0.24, 1), (performance, 0.21, 1), (power, 0.17, 1), (maximum, 0.17, 1), (otto, 0.12, 1), (carnot, 0.12, 1), (refrigerators, 0.1, 1)

(granular, 0.85, 2), (gas, 0.17, 1), (gases, 0.16, 1), (inelastic, 0.13, 1), (driven, 0.13, 1)

(chimera, 0.7, 1), (states, 0.35, 1), (oscillators, 0.33, 1), (coupled, 0.31, 2), (networks, 0.2, 3), (nonlocally, 0.13, 1), (chimeras, 0.12, 1), (coupling, 0.1, 1)]

(dynamics, 0.19, 1), (model, 0.18, 5), (networks, 0.17, 3), (liquid, 0.16, 4), (diffusion, 0.13, 2), (phase, 0.13, 4), (quantum, 0.13, 3), (dimensional, 0.12, 1), (random, 0.12, 2), (flow, 0.11, 2), (systems, 0.11, 1), (plasma, 0.11, 1), (coupled, 0.1, 2), (time, 0.1, 1)

(dynamic, 0.41, 1), (ising, 0.35, 2), (phase, 0.34, 4), (oscillating, 0.34, 1), (field, 0.32, 3), (transition, 0.24, 1), (kinetic, 0.2, 1), (model, 0.2, 5), (magnetic, 0.15, 1), (nonequilibrium, 0.13, 1), (blume, 0.12, 1), (capel, 0.12, 1), (transitions, 0.11, 1)

(biaxial, 0.53, 1), (nematic, 0.5, 2), (liquid, 0.29, 4), (crystals, 0.19, 4), (phases, 0.19, 2), (bent, 0.17, 1), (phase, 0.16, 4), (molecules, 0.15, 1), (core, 0.14, 1), (simulation, 0.12, 1), (molecular, 0.1, 1), (antinematic, 0.1, 1), (mesogenic, 0.1, 1)

(passive, 0.47, 1), (scalar, 0.41, 1), (anomalous, 0.39, 1), (scaling, 0.29, 1), (advected, 0.24, 1), (turbulence, 0.22, 1), (turbulent, 0.18, 1), (advection, 0.15, 1), (loop, 0.12, 1), (anisotropy, 0.11, 1), (anisotropic, 0.11, 1), (renormalization, 0.11, 1), (vector, 0.11, 1), (field, 0.1, 3)

(quantum, 0.51, 3), (decay, 0.45, 1), (loschmidt, 0.33, 1), (echo, 0.33, 1), (fidelity, 0.25, 1), (chaotic, 0.23, 1), (semiclassical, 0.18, 1), (lyapunov, 0.13, 1), (perturbations, 0.11, 1)

(casimir, 0.69, 1), (critical, 0.37, 1), (forces, 0.27, 1), (films, 0.13, 1), (size, 0.13, 1), (force, 0.13, 1), (finite, 0.12, 1), (free, 0.11, 1), (ising, 0.11, 2), (thermodynamic, 0.1, 1), (model, 0.1, 5)

(traffic, 0.88, 1), (flow, 0.3, 2), (model, 0.13, 5), (car, 0.13, 1), (following, 0.11, 1)

(rogue, 0.62, 1), (schrödinger, 0.34, 1), (waves, 0.31, 2), (wave, 0.29, 2), (equation, 0.25, 1), (nonlinear, 0.21, 2), (solutions, 0.17, 1), (soliton, 0.12, 1), (solitons, 0.11, 1)

(cooperation, 0.6, 1), (dilemma, 0.38, 1), (prisoner, 0.34, 1), (game, 0.25, 1), (games, 0.24, 2), (evolutionary, 0.19, 1), (networks, 0.18, 3), (spatial, 0.17, 1), (social, 0.14, 1), (public, 0.12, 1), (goods, 0.1, 1)

(granular, 0.59, 2), (chains, 0.36, 1), (chain, 0.32, 1), (propagation, 0.22, 1), (waves, 0.21, 2), (nonlinear, 0.2, 2), (solitary, 0.2, 1), (wave, 0.17, 2), (pulse, 0.15, 1), (crystals, 0.14, 4), (strongly, 0.12, 1)

(b) Feature: node neighborhood size

(relation, 0.37, 1), (entropy, 0.34, 1), (differences, 0.34, 2), (production, 0.34, 1), (theorem, 0.34, 1), (work, 0.31, 2), (fluctuation, 0.29, 1), (nonequilibrium, 0.27, 2), (free, 0.27, 5), (energy, 0.27, 3)

(model, 0.25, 5), (phase, 0.23, 5), (dimensional, 0.2, 3), (dynamics, 0.18, 6), (time, 0.17, 4), (networks, 0.16, 10), (lattice, 0.15, 7), (systems, 0.15, 8), (granular, 0.13, 6), (stochastic, 0.13, 2), (random, 0.12, 6), (noise, 0.12, 1), (liquid, 0.12, 4), (nonlinear, 0.12, 2), (field, 0.12, 2), (diffusion, 0.11, 4), (quantum, 0.11, 4), (coupled, 0.11, 2), (transition, 0.11, 5), (boltzmann, 0.1, 6)

(model, 0.28, 5), (networks, 0.25, 10), (lattice, 0.22, 7), (boltzmann, 0.21, 6), (equations, 0.19, 2), (stochastic, 0.15, 2), (dynamics, 0.14, 6), (transition, 0.13, 5), (synchronization, 0.13, 3), (granular, 0.13, 6), (time, 0.13, 4), (scale, 0.13, 3), (glass, 0.13, 2), (systems, 0.12, 8), (random, 0.12, 6), (dimensional, 0.12, 3), (phase, 0.12, 5), (diffusion, 0.11, 4), (complex, 0.1, 3), (reaction, 0.1, 1), (free, 0.1, 5)

(model, 0.32, 5), (microstates, 0.27, 1), (auxiliary, 0.27, 1), (violating, 0.27, 1), (connections, 0.24, 1), (generate, 0.24, 1), (steady, 0.22, 1), (collisions, 0.22, 1), (ising, 0.2, 1), (approach, 0.2, 2), (distribution, 0.2, 1), (second, 0.2, 1), (law, 0.2, 1), (generalized, 0.2, 1), (arbitrary, 0.2, 2), (equilibrium, 0.18, 2), (synchronization, 0.18, 3), (chaos, 0.17, 1), (gases, 0.17, 2), (states, 0.17, 2), (granular, 0.15, 6), (networks, 0.13, 10)

(equation, 0.22, 4), (fokker, 0.19, 1), (planck, 0.19, 1), (hard, 0.19, 2), (fractional, 0.17, 1), (dynamics, 0.16, 6), (observable, 0.13, 1), (evolution, 0.12, 1), (characteristics, 0.12, 1), (quasistatic, 0.12, 1), (correction, 0.12, 1), (cohesion, 0.12, 1), (pair, 0.12, 1), (nearly, 0.12, 1), (ordered, 0.12, 1), (characterization, 0.12, 1), (preasymptotic, 0.12, 1), (formulas, 0.12, 1), (thermalization, 0.12, 1), (depinning, 0.12, 1), (theorems, 0.11, 1), (low, 0.11, 1), (amorphous, 0.11, 2), (intermittency, 0.11, 1), (hydrodynamics, 0.11, 1), (avalanche, 0.11, 1), (athermal, 0.11, 1), (correlation, 0.11, 2), (transport, 0.11, 1), (solution, 0.11, 1), (jammed, 0.11, 2), (propelled, 0.11, 1), (collective, 0.11, 1), (interacting, 0.11, 1), (asymptotic, 0.11, 1), (heterogeneity, 0.11, 1), (singularities, 0.11, 1), (dense, 0.1, 2), (highly, 0.1, 1), (near, 0.1, 1), (inelastic, 0.1, 1), (quantum, 0.1, 4), (fluid, 0.1, 2), (self, 0.1, 2), (shear, 0.1, 2), (rheology, 0.1, 2), (flow, 0.1, 3), (work, 0.1, 2), (liquids, 0.1, 1), (growth, 0.1, 1), (laws, 0.1, 1), (application, 0.1, 1), (disordered, 0.1, 1), (walks, 0.1, 1)

(networks, 0.38, 10), (scientific, 0.22, 1), (collaboration, 0.19, 1), (path, 0.19, 1), (ii, 0.16, 1), (density, 0.14, 2), (diffusion, 0.14, 4), (herds, 0.12, 1), (theory, 0.12, 4), (systems, 0.12, 8), (granular, 0.12, 6), (random, 0.12, 6), (schools, 0.11, 1)

(lattice, 0.27, 7), (networks, 0.26, 10), (boltzmann, 0.21, 6), (phase, 0.18, 5), (models, 0.16, 1), (structure, 0.16, 2), (method, 0.16, 4), (interactions, 0.14, 1), (self, 0.14, 2), (network, 0.13, 2), (community, 0.13, 2), (social, 0.12, 1), (free, 0.12, 5), (dimensions, 0.12, 1), (scale, 0.12, 3), (granular, 0.11, 6), (random, 0.11, 6), (systems, 0.11, 8), (motion, 0.1, 1), (graphs, 0.1, 2), (transition, 0.1, 5), (emulsions, 0.1, 1)

(networks, 0.24, 10), (glass, 0.17, 2), (transition, 0.17, 5), (lattice, 0.15, 7), (solutions, 0.14, 1), (systems, 0.14, 8), (equations, 0.13, 2), (lévy, 0.13, 1), (large, 0.13, 1), (external, 0.13, 1), (jammed, 0.13, 2), (flights, 0.13, 1), (correlated, 0.13, 1), (quantum, 0.12, 4), (coupled, 0.12, 2), (analysis, 0.12, 1), (force, 0.12, 1), (colloidal, 0.12, 1), (order, 0.12, 1), (packings, 0.11, 1), (synchronization, 0.11, 3), (hard, 0.11, 2), (time, 0.11, 4), (langevin, 0.11, 1), (fluctuations, 0.11, 1), (density, 0.1, 2)

(model, 0.24, 5), (dynamics, 0.22, 6), (dimensional, 0.2, 3), (phase, 0.19, 5), (liquid, 0.15, 4), (nonlinear, 0.15, 2), (systems, 0.14, 8), (field, 0.13, 2), (time, 0.12, 4), (quantum, 0.11, 4), (transition, 0.11, 5), (diffusion, 0.11, 4), (flow, 0.1, 3), (induced, 0.1, 1)

(networks, 0.41, 10), (evaluating, 0.36, 1), (uncorrelated, 0.36, 1), (generation, 0.36, 1), (finding, 0.3, 1), (structure, 0.28, 2), (community, 0.28, 2), (free, 0.26, 5), (scale, 0.26, 3), (random, 0.23, 6)

(zero, 0.55, 1), (epitome, 0.3, 1), (applied, 0.27, 1), (applications, 0.27, 1), (distributions, 0.25, 1), (stress, 0.25, 1), (arbitrary, 0.23, 2), (disorder, 0.23, 1), (degree, 0.23, 1), (temperature, 0.23, 1), (jamming, 0.23, 1), (graphs, 0.21, 2), (random, 0.17, 6)

(measurements, 0.35, 1), (hierarchical, 0.32, 1), (differences, 0.32, 2), (approach, 0.3, 2), (organization, 0.3, 2), (master, 0.3, 1), (equilibrium, 0.27, 2), (nonequilibrium, 0.25, 2), (free, 0.25, 5), (energy, 0.25, 3), (complex, 0.24, 3), (equation, 0.21, 4), (networks, 0.2, 10)

(nucleotides, 0.29, 1), (hove, 0.29, 1), (mixing, 0.26, 1), (van, 0.26, 1), (mosaic, 0.26, 1), (correlation, 0.24, 2), (dna, 0.24, 1), (patterns, 0.24, 1), (organization, 0.22, 2), (testing, 0.22, 1), (mixture, 0.22, 1), (lennard, 0.2, 1), (jones, 0.2, 1), (supercooled, 0.2, 1), (function, 0.19, 1), (coupling, 0.19, 1), (mode, 0.19, 1), (binary, 0.19, 2), (theory, 0.17, 4), (networks, 0.15, 10)

(lattice, 0.42, 7), (percolation, 0.32, 1), (boltzmann, 0.26, 6), (monte, 0.21, 1), (term, 0.21, 1), (carlo, 0.21, 1), (site, 0.19, 1), (forcing, 0.19, 1), (fast, 0.17, 1), (bond, 0.17, 1), (transitions, 0.17, 1), (algorithm, 0.17, 1), (effects, 0.16, 1), (liquid, 0.16, 4), (phase, 0.16, 5), (discrete, 0.16, 2), (network, 0.16, 2), (gas, 0.16, 2), (simulation, 0.16, 2), (nonideal, 0.16, 1), (small, 0.15, 1), (world, 0.15, 1), (scaling, 0.15, 2), (gases, 0.15, 2), (model, 0.14, 5), (method, 0.14, 4), (equation, 0.12, 4)

(viscoplastic, 0.33, 1), (dissipation, 0.29, 1), (deformation, 0.29, 1), (isotropy, 0.29, 1), (dispersion, 0.29, 1), (amorphous, 0.27, 2), (stability, 0.27, 1), (solids, 0.27, 1), (galilean, 0.27, 1), (invariance, 0.27, 1), (dynamics, 0.2, 6), (method, 0.2, 4), (lattice, 0.2, 7), (boltzmann, 0.19, 6), (theory, 0.19, 4)

(boltzmann, 0.63, 6), (lattice, 0.5, 7), (equation, 0.29, 4), (simulations, 0.21, 1), (fluid, 0.21, 2), (liquid, 0.19, 4), (gas, 0.19, 2), (binary, 0.18, 2), (method, 0.17, 4), (theory, 0.16, 4), (systems, 0.16, 8)

(plane, 0.38, 1), (flow, 0.32, 3), (dynamics, 0.26, 6), (granular, 0.24, 6), (rheophysics, 0.24, 1), (endemic, 0.21, 1), (temperatures, 0.19, 1), (equilibrium, 0.19, 1), (bagnold, 0.19, 1), (inclined, 0.19, 1), (partial, 0.17, 1), (effective, 0.17, 1), (flows, 0.16, 1), (dense, 0.16, 2), (epidemic, 0.16, 1), (shear, 0.16, 2), (slow, 0.16, 1), (rheology, 0.16, 2), (discrete, 0.15, 2), (simulation, 0.15, 2), (states, 0.14, 2), (scaling, 0.14, 2), (materials, 0.14, 1), (energy, 0.14, 3), (complex, 0.13, 3), (systems, 0.12, 8), (networks, 0.11, 10)

