# OpenReview forum: "Ollivier-Ricci Curvature for Hypergraphs: A Unified Framework"
_ICLR.cc/2023/Conference — ICLR 2023 poster_

### Official Review · Reviewer_LgSf · 2022-10-20

**Confidence:** 3
**Correctness:** 4
**Technical Novelty And Significance:** 3
**Empirical Novelty And Significance:** 3
**Recommendation:** 6

**Clarity, Quality, Novelty And Reproducibility:**

The paper is well drafted.I think the innovation is good, and it is the first few papers to generalize curvature to hyper-graphs.

**Strength And Weaknesses:**

strength:  1. they use several ways to generalize graph Ricci curvature into hypergraphs based on general features of hypergraphs.
2. They derive the bound of the newly defined curvatures.  3. using the newly defined curvatures to show its empirical advantanges compare to single node information.

**Summary Of The Paper:**

This paper proposes a unified framework for Ollivier-Ricci curvature on hyper-graphs, by incorporating and generalising existing approaches to hyper-graphs. It can be regarded as the first attempt towards defining RC curvature, based on rigorous theoretical and empirical analysis of the resulting curvature formulation.

**Summary Of The Review:**

The paper is in excellent shape and worth for an acceptance.

---

> ### Author Response · Authors · 2022-11-10
> **Response to Reviewer LgSf**
>
> Thank you for your review, and for acknowledging that our paper is worthy of acceptance. We hope that the changes we made in response to other reviewers’ comments further improved our submission. As your current recommendation places our paper “marginally above the acceptance threshold”: Is there anything else we can do to strengthen our case?

---

### Official Review · Reviewer_Bvqz · 2022-10-21

**Confidence:** 2
**Correctness:** 4
**Technical Novelty And Significance:** 4
**Empirical Novelty And Significance:** 4
**Recommendation:** 8

**Clarity, Quality, Novelty And Reproducibility:**

The paper is well written. The contributions seem to be quite novel. All the codes and data is made publicly available.

**Strength And Weaknesses:**

The paper is quite dense (especially for a non-expert in graph theory) but well-written. It seems that the proposed notions are diverse and maybe separately important in various applications. The experimental validation seems to be appropriate.

What is missing is that how other curvature measures on hypergraphs, such as FRCs, compare to the proposed extension of ORCs. Since the framework does not extend FRCs, it might have been interesting to see how differently it act, if that is the case.


**Summary Of The Paper:**

The paper provides a framework for computing Olivier-Ricci curvature (ORC) on hypergraphs. In order to extend ORCs to hypergraphs, it provides multiple notions of probability measures and Similarity measures (extension to Wasserstein distance in graphs). The non-redundancy of these notions is shown through experiments on data from various domains.

**Summary Of The Review:**

The novelty of the contributions overweigh the minor weaknesses.

---

> ### Author Response · Authors · 2022-11-10
> **Response to Reviewer Bvqz**
>
> Thank you for your review, and for advocating the acceptance of our paper.
>
> Regarding the comparison with other hypergraph curvature measures such as hypergraph FRC, we fully agree that this is an interesting direction, as indicated toward the end of our conclusion (Section 6 (“Discussion and Conclusion”)). But while there exist a couple of FRC definitions for hypergraphs in the literature (cf. Section 4 (“Related Work”)), these have not yet been integrated into a framework that would allow their systematic exploration. We consider the development of such a framework to be the natural next step, after which we will be able to investigate thoroughly both the relationships between ORC and FRC curvatures on hypergraphs and the role of the individual hypergraph-specific turning knobs in shaping these relationships.
>
> Finally, if there is anything we can do to improve the digestibility of our paper for non-experts in graph theory within the constraints of the page limits (or as addenda in the Appendix), or to increase your confidence in your paper recommendation, please let us know, as we will readily attempt to accommodate your feedback.

---

### Official Review · Reviewer_x9Zy · 2022-10-24

**Confidence:** 3
**Correctness:** 3
**Technical Novelty And Significance:** 3
**Empirical Novelty And Significance:** 3
**Recommendation:** 5

**Clarity, Quality, Novelty And Reproducibility:**

The paper focuses on analyzing the three random walks of Equal-Nodes, Equal-Edges, and Weighted-Edges, but many higher-order correlations in the hypergraph cannot be generalized by these three methods. Many details in the paper need to be improved, such as the definition and expression of the formulas, the information in the figures, etc. This article is the first to generalize the Ollivier-Ricci curvature of graphs to hypergraphs, which is original. A rigorous and clear proof of the theorem is given in the paper.

**Strength And Weaknesses:**

Strength:
This paper analyzes three probability measures based on random walks on hypergraphs. The related formulas and definitions of ORC in graphs are extended to hypergraphs, making their concepts consistent with geometric intuitions and rigorous proofs.

Weaknesses:
1. Figure 1 needs to be more clear and intuitive. For example, too many directed edges are used in Figure 1(d) to describe the random walk of weighted edges. It is recommended to use the relationship between the weighted edge and the hyperedge cardinality to represent, so that would be more concise.
2. The definition of AGG(s) in Eq. (14) is ambiguous. In this paper, it is said that Eq.(14) is consistent with the previously defined hyperedge curvature. But should we understand it as the sum of AGG for nodes in subset s or  understand it as AGGM(s) ?
3. In Corollary6, only Equal-Nodes Random Walk is analyzed, but Equal-Edges Random Walk and Weighted-Edges Random Walk are not analyzed.
4. In Figure 3a, the shapes and colors in the legend indicate errors, languages should be represented by upper triangles, and Nerds should be represented by dots.
5. This paper uses unlabeled datasets for experiments and then uses the Wasserstein Clustering Coefficient to measure the results. We recommend using a labeled dataset, so that ground truth can be accurately obtained, allowing for more intuitive comparison of results.
6. This paper generalizes the Ollivier-Ricci curvature of graphs to hypergraphs, and the experimental part should focus on the advantages of using hypergraph curvature and compare ORCHID with methods which using Ollivier-Ricci curvature for graphs.

**Summary Of The Paper:**

This paper introduces a unified framework called Ollivier-Ricci Curvature for Hypergraphs In Data (ORCHID). This framework is the first to generalize the Ollivier-Ricci curvature (ORC) of graphs to hypergraphs and yields a notion of curvature consistent with geometric intuition. This work provides rigorous theoretical and empirical analysis of the resulting curvature formulations and demonstrates the utility of these concepts in practice through extensive experiments.

**Summary Of The Review:**

The sufficiency and necessity of the paper are the important part and generalization from graphs to hypergraphs should focus on considering higher-order associations of hypergraphs, as well as highlighting the advantages compared to graph-based frameworks.

---

> ### Author Response · Authors · 2022-11-10
> **Response to Reviewer x9Zy (Part 1/2)**
>
> Thank you for your review, and for recognizing the originality of our approach. In the following, we address each of your critiques in turn.
> ### Numbered Weaknesses
> 1. Figure 1: In Figures 1b–d, as explained in the caption, we draw arrows such that all arrows outgoing from the same node or hyperedge can be traversed with uniform probability. If we implement your suggestion to modify Figure 1d, this intuitive interpretation is lost (note that “weighted-edges random walk” is the name we assign to the walk because it _effectively_ weights hyperedges by their cardinality, not because the hyperedges are weighted to start with). Additionally, we run into the problem that we need to encode weights/arrow traversal probabilities, which can be done via arrow annotations (messy) or line thickness (hard to read). We have uploaded Figure 1d drawn using the original and alternative encodings in [this anonymous Google Drive folder](https://drive.google.com/drive/folders/1kYBmyrQezHY7XJYZEY5e6erBMURFadMW?usp=sharing) for your reference, and we will happily include the version you deem most suitable in our revised manuscript.
> 2. Eq. (14): The “ambiguity” you identified is on purpose, as you can simply plug in any aggregation function of your choosing, including AGG_A, AGG_B, or AGG_M. Note that we always define AGG for a set of nodes (undirected hyperedges are sets of nodes, and so is any subset of the node set), and hence, AGG(s) for s ⊆ V should be read as “applying a set aggregation function such as AGG_A, AGG_B, or AGG_M to the set s ⊆ V”.  We amended the language in the manuscript to clarify this.
> 3. Corollary 6: It is deliberate that we analyze “only” the equal-nodes random walk here, as the purpose of Corollary 6 is to establish the relationship between ORC for graphs and ORCHID for hypergraphs, and the two just so happen to be connected via the equal-nodes random walk; see also our response to point 6 (subpoint c). We amended the language in the manuscript to clarify this.
> 4. Legend of Figure 3a: Thank you for catching the marker mixup in the legend. We have fixed this in the revised version.
> 5. Labeled datasets: While aiming at ground truth might be highly useful in some settings, it is rather problematic in (many) others. Most importantly, we need reliable labels that we want to predict/respect _no matter what_, i.e., it must be reasonable to assume that what we want to tease out of the data using our method is, fundamentally, already known. This is not the case in our setting for two reasons. First, and most importantly, our main contribution is a systematic framework for generalizing Ollivier-Ricci curvature to hypergraphs that yields descriptors which tease out interesting structure in the data that is _not_ already known (which we recognize makes it _much_ harder to evaluate our method). Second, we are not aware of any hypergraph dataset with high-quality, natural ground truth that would be suitable for our experiments (in the sense that one could reasonably expect Ollivier-Ricci curvature to recover the labels). Did you have any specific datasets in mind here? Finally, note that in Appendix A.5 under Q3 (Hypergraph Learning), we use the titles of Physical Review E articles to ascertain that the clusterings we obtain are semantically meaningful. This makes our results interpretable (given some background knowledge in physics) even without treating anything as ground truth.
> 6. Focus of the experimental part: We agree that it is important to understand how ORCHID compares with (and is preferable over) graph ORC. However, we do not need to do this experimentally (and hence, can explore other aspects of ORCHID in our experiments) because we already know the relevant relationships from our theoretical analyses. We now explicitly state this at the beginning of Section 5 (“Experiments”), and amended the language around the relevant results to clarify the connection. Summarizing the key points:
>
>     a. When applied to graphs, ORCHID simplifies to graph ORC (Lemmas 1 and 2 plus the sentence that follows to clarify this implication).
>
>     b. By Corollary 6, ORCHID with an equal-nodes random walk is equivalent to ORC on graphs representing hypergraphs by their clique expansion, which is how most graph datasets out there implicitly model higher-order relations.
>
>     c. By construction, equal-edges and weighted-edges random walks capture more information, and hence, are more powerful (e.g., for distinguishing non-isomorphic hypergraphs) than the equal-nodes random walks from which we can (re)construct graph ORC.

---

> > ### Author Response · Authors · 2022-11-10
> > **Response to Reviewer x9Zy (Part 2/2)**
> >
> > ### Random Walks and Higher-Order Correlations/Associations
> > You rightfully observe that the random walks we focus on in the paper do not incorporate higher-order correlations/associations, but our framework can easily account for such associations by using multi-hop random walks in the definition of its probability measure (i.e., spreading probability mass to neighbors beyond distance 1). As ORC on graphs was originally introduced using one-hop random walks, we focused on this case when introducing our framework. However, _given_ our framework, it is now clear how higher-order correlations can be integrated into ORC-like measures for hypergraphs, and we see exploring such measures as a promising direction for future work (see Section 6 (“Discussion and Conclusion”)).
> > ### Clarity
> > > Many details in the paper need to be improved, such as the definition and expression of the formulas, the information in the figures, etc.
> >
> > As you were the only reviewer to find our submission partially unclear in these respects: Is there anything particular, beyond the points you raised in the “Weaknesses” section (addressed above), that we can do to improve your reading experience?
> > ### Correctness
> > Your rating currently indicates that “Some of the paper’s claims have minor issues. A few statements are not well-supported, or require small changes to be made correct.” Is this fixed with our responses to your questions regarding Eq. (14) and Corollary 6 (explained above), or is there something else you would like us to clarify?
> >
> > Again, thank you for your feedback on our manuscript. Should we have addressed your remaining concerns, we would be grateful if you considered reflecting this in your score.

---

### Official Review · Reviewer_22tj · 2022-10-26

**Confidence:** 3
**Correctness:** 4
**Technical Novelty And Significance:** 3
**Empirical Novelty And Significance:** 3
**Recommendation:** 8

**Clarity, Quality, Novelty And Reproducibility:**

*Clarity*

I found the paper easy to read and parse. The things that were not clear in my mind seem to be limitations of the paper itself.

*Novelty*

The present approach to curvature seems genuinely novel.

*Quality*

The results of the article are quite intriguing, but I feel they are somewhat incomplete. To my mind, the most compelling empirical results are the ones in Figure 3 and Table 2, which allow for quick comparisons of different features.

I think a few other directions could be discussed with respect to the exploratory analysis. For instance, what is the geometric meaning of curvature in examples (eg. are highly cited articles extremal for curvature because the literature "diverges" in many different directions away from them?).

*Reproducibility*

I did not check this carefully but it seems possible to reproduce everything with the material given.

**Strength And Weaknesses:**

*Strengths*

- The paper presents a principled method for defining curvature parameters in hypergraphs.
- Some mathematical results and properties are presented.
- These new parameters seem to serve as good features for clustering and other tasks, as demonstrated in an experimental study. Figure 3 and Table 2 are compelling.

*Weaknesses*

- Although I am happy with some experiments, I found it hard to draw conclusions from the experiments in general. For instance, I do not understand to what the authors are comparing Table 13 to.
- More broadly, since are other notions of curvature for hypergraphs, and other methods for extracting hypergraph features, I would like to understand how the new notion introduced here compares to the alternatives in practice. What is the state of the art?

**Summary Of The Paper:**

The paper introduces a notion of curvature on hypergraphs. It is related to Ollivier's notion, which works for metric spaces with Markov chains on them. In that case, curvature is (basically) the curvature describes the contraction of the map from points x to the transition probabilities from x.

There is a "canonical" choice of Markov chain for graphs -- (lazy) random walk -- but there's no obvious choice for hypergraphs, so the paper explores three possibilities. It also proves some bounds and intuitive properties of these definitions.

One motivation for introducing a new notion of curvature is to have a "local" (and thus simple to compute) parameter that is indicative of "global" features of the network. This idea is tested in an exploratory empirical study, where the newly introduced curvatures seem to function as meaninful features for clustering and hypergraph exploration/node differentiation.

**Summary Of The Review:**

The paper offers an intriguing new notion of curvature for hypergraphs that seems to perform well in practice. My main misgivings are about the lack of more systematic comparisons with other approaches.

--

UPDATE on Nov 30th:

I am a bit more positive about the comparison with other approaches, and have raised my score accordingly.

---

> ### Author Response · Authors · 2022-11-10
> **Response to Reviewer 22tj**
>
> Thank you for your review, and for appreciating the novelty of our approach to hypergraph curvature. In the following, we address the critiques you raised in your comments.
> ### Conclusions from the Experiments
> We are glad that you found the results of our paper intriguing, and we share the sentiment that there remains a lot to investigate (in future work). To us, the systematic generalization of graph ORC to hypergraphs in our ORCHID framework is the main contribution of our current submission, and we establish already theoretically that ORCHID strictly generalizes both ORC on graphs and existing approaches to hypergraph ORC. The main goal of our experiments, then, is to show that the resulting curvature notions are meaningful in practice; we do not aim to show that ORCHID curvatures outperform highly engineered machine learning models at specific tasks. For each figure or table (excluding dataset details in Appendix A.3), we give the “punchline” as the first sentence of its caption. In the original manuscript, Table 13 only serves to show that ORCHID produces semantically coherent clusterings; the full results (based on 28 clusterings using ORCHID and non-ORCHID features and different clustering methods) are included in the reproducibility material. In response to the first bullet point you stated under “Weaknesses”, we now display two subtables in Table 13: the original Table 13 as Table 13a (where we corrected the accidental omission of one row), and a comparable table for a clustering result based on non-ORCHID features as Table 13b.
> ### Comparisons with other Approaches
> As detailed in Section 4 ("Related Work"), our framework subsumes existing approaches to ORC on hypergraphs, which means that we cover these approaches experimentally by exploring ORCHID curvature notions.
>
> Regarding the comparison with _other hypergraph curvature measures_ such as hypergraph FRC, we fully agree that this is an interesting direction, as indicated toward the end of our conclusion. But while there exist a couple of FRC definitions for hypergraphs in the literature (cf. Section 4), these have not yet been integrated into a framework that would allow their systematic exploration. We consider the development of such a framework to be the natural next step, after which we will be able to investigate thoroughly both the relationships between ORC and FRC curvatures on hypergraphs and the role of the individual, hypergraph-specific turning knobs in shaping these relationships.
>
> As far as _other hypergraph features_ are concerned, we chose _four_ general-purpose features as competitors in our experiments to show that ORCHID curvatures provide more nuance than other general-purpose features of similar locality and complexity, and we welcome further suggestions regarding such features to include in the final version of our manuscript. We deliberately refrain from doing any experiments requiring hypergraph data with ground truth for two reasons. First, and most importantly, our main contribution is a systematic framework for generalizing Ollivier-Ricci curvature to hypergraphs that yields descriptors which tease out interesting structure in the data that is _not_ already known (which we recognize makes it _much_ harder to evaluate our method). Second, we are not aware of any hypergraph dataset with high-quality, natural ground truth that would be suitable for our experiments (in the sense that one could reasonably expect Ollivier-Ricci curvature to recover the labels).
> ### Other Directions for Exploratory Analysis
> Thank you for your interest in our exploratory analysis, and for your suggestions to extend it. We now give more details on the interpretation of our exploratory results, including the geometric meaning of curvature in our examples, in Appendix A.5 under Q2 (“Hypergraph Exploration”).
>
> Again, thank you for your feedback on our manuscript. Should you be satisfied with our responses to your concerns, we would be grateful if you considered reflecting this in your score.

---

### Author Response · Authors · 2022-11-10
**General Response to Reviewers**

We thank all reviewers for their comments, and for the largely positive evaluation of our submission. In our responses to the individual reviews, we address each reviewer’s remaining critiques. For your convenience, we highlight all substantial changes as green text in the revised manuscript.

---

### Author Response · Authors · 2022-11-17
**Happy to provide further clarifications!**

We thank the reviewers for their positive view of our work! In light of the approaching deadline for the rebuttal (Nov 18), we would like to ask whether we may provide any additional clarifications. Moreover, if you find our answers to your comments satisfactory, we would appreciate if this was reflected in your score.

---

### Decision · Program_Chairs · 2023-01-20

**Decision:**

Accept: poster

**Justification For Why Not Higher Score:**

While the experiments show that the notion of hypergraph curvature is meaningful in practice, it is not yet clear what the takeaway message is or how and when to use it. Also the results could be more compelling with further comparisons to FRCs.

**Justification For Why Not Lower Score:**

The paper takes aim at an ambitious question of defining curvature for hypergraphs. It lays important theoretical foundation, and makes a compelling case that it is a useful tool to have for some hypergraph applications.

**Metareview: Summary, Strengths And Weaknesses:**

Curvature plays a key role in geometry, probability and optimization. However thus far it has been limited to the graph or manifold setting. This paper explores the notion of curvature in hypergraphs. In the graph setting, curvature measures the contraction of the map from points x to its transition probabilities. For hypergraphs there are many natural ways to define an associated markov chain, and this paper systematically explores them. Furthermore through real and synthetic experiments, they demonstrate how their definition of curvature can both be computed at scale and is useful for a variety of tasks.

**Note From Pc:**

if the above contains the word "oral" or "spotlight" please see: "oral" presentation means -> notable-top-5% and "spotlight" means -> notable-top-25%. As stated in our emails, we are disassociating presentation type from AC recommendations

**Summary Of Ac-Reviewer Meeting:**

N/A